# ProTInSeq: transposon insertion tracking by ultra-deep DNA sequencing to identify translated large and small ORFs

Samuel Miravet-Verde[1,6] ✉, Rocco Mazzolini[2], Carolina Segura-Morales [1], Alicia Broto [1], Maria Lluch-Senar[2,3] ✉ & Luis Serrano [1,4,5] ✉

Identifying open reading frames (ORFs) being translated is not a trivial task. ProTInSeq is a technique designed to characterize proteomes by sequencing transposon insertions engineered to express a selection marker when they occur in-frame within a protein-coding gene. In the bacterium *Mycoplasma pneumoniae*, ProTInSeq identifies 83% of its annotated proteins, along with 5 proteins and 153 small ORF-encoded proteins (SEPs; ≤100 aa) that were not previously annotated. Moreover, ProTInSeq can be utilized for detecting translational noise, as well as for relative quantification and transmembrane topology estimation of fitness and non-essential proteins. By integrating various identification approaches, the number of initially annotated SEPs in this bacterium increases from 27 to 329, with a quarter of them predicted to possess antimicrobial potential. Herein, we describe a methodology complementary to Ribo-Seq and mass spectroscopy that can identify SEPs while providing other insights in a proteome with a flexible and cost-effective DNA ultra-deep sequencing approach.

The genome annotation consortia relies on 100 and 50 amino acid (aa) cutoffs to distinguish protein-coding sequences in eukaryotic and prokaryotic genomes, respectively[1–3]. However, integrative analyses have revealed the existence of translated, as-of-yet non-annotated, proteins of ≤100 aa encoded by small open reading frames (smORFs), termed SEPs (from smORF-encoded proteins). SEPs are often excluded in the automated prediction of protein-coding open reading frames (ORFs) to counteract false positive prediction[4]. Thus, SEPs represent a large pool of poorly understood molecules despite having central roles in sporulation[5], influx inhibition[6], photosynthesis[7], cell division, stress sensing, and antibiotic resistance[3], assembly of mitochondrial respiratory complexes[8], host inflammation and immunity[9]. Although different approaches primarily based on computational predictions identify some of them[10–16], experimental high-throughput

methodologies to validate and characterize SEPs are hindered by several barriers[17]. Techniques based on RNA-seq (such as ribosome profiling by Ribo-Seq, which sequences fragments of transcripts bound by ribosomes) can be effective at detecting translated smORFs[12,18]. For example, meta-transcriptomics analyses of microbiome genomes reveal >4,000 bacterial SEP families and ~40,000 in phages[19], 30% of these predicted to be secreted and/or transmembrane[20]. However, in bacteria, transcriptional units are polycistronic, and a vast number of smORFs overlap with longer ORFs, making their identification by this approach challenging. The product of a translated smORF can be either a functional SEP, such as communication and competition SEPs in microbial communities[21,22], or translational noise. Nevertheless, this noise can act as an evolutionary reservoir to more complex proteins[23], mitigate transcriptional noise to

[1]Centre for Genomic Regulation (CRG), The Barcelona Institute of Science and Technology, Dr Aiguader 88, 08003 Barcelona, Spain. [2]Pulmobiotics, Dr Aiguader 88, 08003 Barcelona, Spain. [3]Institute of Biotechnology and Biomedicine "Vicent Villar Palasi" (IBB), Universitat Autònoma de Barcelona, Barcelona, Spain. [4]Universitat Pompeu Fabra (UPF), Barcelona, Spain. [5]ICREA, Pg. Lluis Companys 23, 08010 Barcelona, Spain. [6]Present address: Department of Biology, Institute of Microbiology and Swiss Institute of Bioinformatics, ETH Zurich, Zurich, Switzerland. ✉e-mail: smiravet@ethz.ch; maria.lluch@pulmobio.com; luis.serrano@crg.eu

achieve more precise protein production[24], or peptides relevant for the bacteria to adapt to different environmental conditions[25]. Experimentally, direct approaches such as 3´-tagging and immunoblotting, which are far from high-throughput, have shown that 36 out of 80 aimlessly selected smORFs in *Escherichia coli* are translated into SEPs[26]. At larger scales, mass spectroscopy (MS) works over precomputed ORF databases that rarely consider smORFs to avoid artifacts[27]. Furthermore, at least two unique tryptic peptides (UTPs) are required for high-confidence protein identification by MS but many SEPs present one or zero UTPs[10]. Thus, new experimental approaches are required to complement Ribo-Seq and MS and validate SEPs in a high-throughput manner.

High-throughput transposon insertion tracking by ultra-deep sequencing (Tn-Seq or HITS) enables genome-wide studies in a varied range of species under diverse conditions with unprecedented depth[28–30]. First, transposon mutagenesis is used to create a library of mutants in which each cell has a transposon inserted into a genomic locus. If insertions span a significant portion of the targeted genome, it becomes possible to identify non-essential (NE, disruptable without compromising cell viability under the studied conditions), fitness-affecting (F, disruptable but affecting cell viability), and essential (E) elements through ultra-deep sequencing[31]. By growing these cells under selective conditions, the prevalence of each mutant proportional to its fitness can be determined because attenuated mutants are outcompeted by those with increased growth and/or survival. Ultra-deep sequencing can quantify transposon insertion sites in the population by sequencing to high depth for specific genomic regions, in this case, transposon insertions and their genomic DNA neighboring regions. Recently, we developed a series of tools to accurately retrieve these profiles[31]. Applications of Tn-Seq methodologies are not limited to essentiality assays[29,32]. For example, transposon-based insertions have been used to produce libraries of fluorescent fusion proteins expressed at endogenous levels for screenings of protein activity and protein localization[33]. In the genome-reduced bacterium *Mycoplasma pneumoniae* (*MPN*); a human lung pathogen that causes atypical pneumonia widely used as a model for Synthetic and Systems Biology[34], Tn-Seq method can reach high coverage with 1 insertion every ~3 bp[31]. Its reduced genome (816,394 bp; 690 annotated proteins) also makes *MPN* a good model for SEPs identification as the number of putative smORFs is less imposing than in other bacterial species. In addition, this organism is particularly sparse in transcriptional and translation regulatory mechanisms as it lacks −35 elements at promoters, producing significant transcriptional noise[35,36]; and it does not need a Ribosome Binding Site (RBS) for the first gene of an operon; rather, translation starts with ribosomes binding to the first start codon found at the 5´ end of mRNA[37,38]. Finally, bioinformatic approaches show this genome-reduced cell could encode for 144 SEPs[10] (including the 27 SEPs already annotated).

In this work, we engineer mini-transposon vectors to identify translated ORFs (including SEPs) in *MPN*. We take the high insertion rate found in its genome into advantage to design Tn4001-derived mini-transposons carrying reporters with no translation initiation codons, so they are expressed only when fused to an endogenous protein (Fig. 1a). We use chloramphenicol acetyltransferase (referred here as '*Cm*')[39] and erythromycin esterase (EreA; referred to as *Ery*) as positive selection markers, and the RNase barnase ('*Barn*') as a negative one[40]. Positive selection requires the insertion of the antibiotic resistance gene in-frame with a protein-coding sequence. Conversely, in negative selection, the *Barn* gene must be out of frame with the genomic protein-coding sequence to ensure the survival of the bacteria. Using this technique, we identify 75% of the annotated *MPN* proteome (80% if we consider the proteome detected by MS), including 24 of its 27 annotated SEPs. Further, we identify 158 non-annotated proteins, of which 153 encoded for SEPs. This technique allows both protein identification and relative quantification of F and NE proteins using different antibiotic concentrations with respect to the protein abundances obtained by MS. Further, it allows the cytoplasmic and the external regions of F and NE transmembrane proteins to be distinguished. Finally, our results strongly suggest that *MPN* lack of RBS regulation entails high levels of translational noise. Integrative analyses with Ribo-Seq and MS show they are similar in terms of detected annotated proteins, with ProTInSeq identifying elusive proteins overlooked by the other experimental methods. In addition, ProTInSeq identifies a notable percentage of these SEPs exhibiting functional potential, as predicted by various computational tools, including high-priority targets like antimicrobials. In summary, we provide with ProTInSeq an orthogonal method to MS and Ribo-Seq to identify SEPs in a bacterial genome, including cases overlapping larger genes, which in addition can reveal quantitative information of a proteome, and validate topological predictions for NE and F membrane proteins. Given the relevant functions described for SEPs and the flexibility and cost-effectiveness of Tn-Seq, we envision ProTInSeq as a valuable tool to validate the expression of elusive SEPs of potential relevance in microbial physiology[41].

## Results
### Mini-transposon engineering to obtain the ProTInSeq library

We used a Tn4001 mini-transposon carrying chloramphenicol acetyltransferase (*Cm*) with an initiating ATG start codon and *Cm* expressed under the control of P$_{438}$ promoter (*i.e.* TnP438CatIR)[29,31]. This mini-transposon vector originally presents stop codons in the three possible frames of the inverted repeats (IR). We mutated the first stop TAA codon to TTA (*CmA* vector), keeping the stop codons in the two alternative ORFs and the P$_{438}$ promoter and starting codon of the *Cm* (Fig. 1b). We then removed a putative −10 Pribnow box in the IR of *CmA* (*CmC* vector; Fig. 1b). The *CmA* and *CmC* vectors were used as positive controls to study whether the transformation efficiency (TE) was affected by the IR mutations (determined by counting colony forming units (CFUs); see Methods). We found no significant changes in the transformation efficiency between the TnP438CatIR original transposon and the *CmA* and the *CmC* vectors (Welch's Test *P* = 0.94 and *P* = 0.41, respectively; Supplementary Data 1). Then we removed the P$_{438}$ promoter and the start codon of the *Cm* resistance from *CmA* and *CmC* vectors to create *CmB* and *CmD*, respectively, so that the marker should only be expressed if inserted in-frame with a translated gene (Fig. 1b). We applied the same rationale to engineer another positive selection vector using an erythromycin-resistance (*Ery*) in two libraries, equivalent to *CmA* (*EryA*; as a control), and to *CmB* (*EryB*; for positive selection). For negative selection, we used a vector with barnase (*Barn*), an RNAse lethal at only a few copies per cell[42], with a constitutively expressed chloramphenicol resistance gene downstream to the *Barn* gene[42–44]. Here, the control vector (*BarnA*) is inherently lethal, whereas the version without a promoter and start codon (*BarnB*) only exhibits lethality when the enzyme is fused in-phase with a gene (Fig. 1c). Positive selector vectors (*CmB*, *CmD* and *EryB*) had lower TE than their respective positive controls (percentage of CFUs in selection relative to controls: 21.9%, 12.5%, and 1.2%, respectively). As expected, the opposite was observed for the barnase, with ten times higher TE in *BarnB* than in *BarnA* (Fig. 1c; Supplementary Data 1).

Using Sanger sequencing, we analyzed 5 individual colonies from the *CmB* library, and 1 from the *CmD* library (Table 1). We identified insertions in 2 NE genes, and at the N- or C-terminal regions in the fitness (F) gene *mpn624* and at the C-terminal region in the E gene *mpn165* (encoding for ribosomal proteins RmpB and RplC, respectively). Notably, another insertion was found in *MPNsO2*, which we previously showed encodes a SEP of 12 aa[10,29]. In all cases, the insertion sites were in-frame with an ORF. Thus, this technique can also be used to study E and F proteins if the insertion site occurs at NE termini regions in a gene[31].

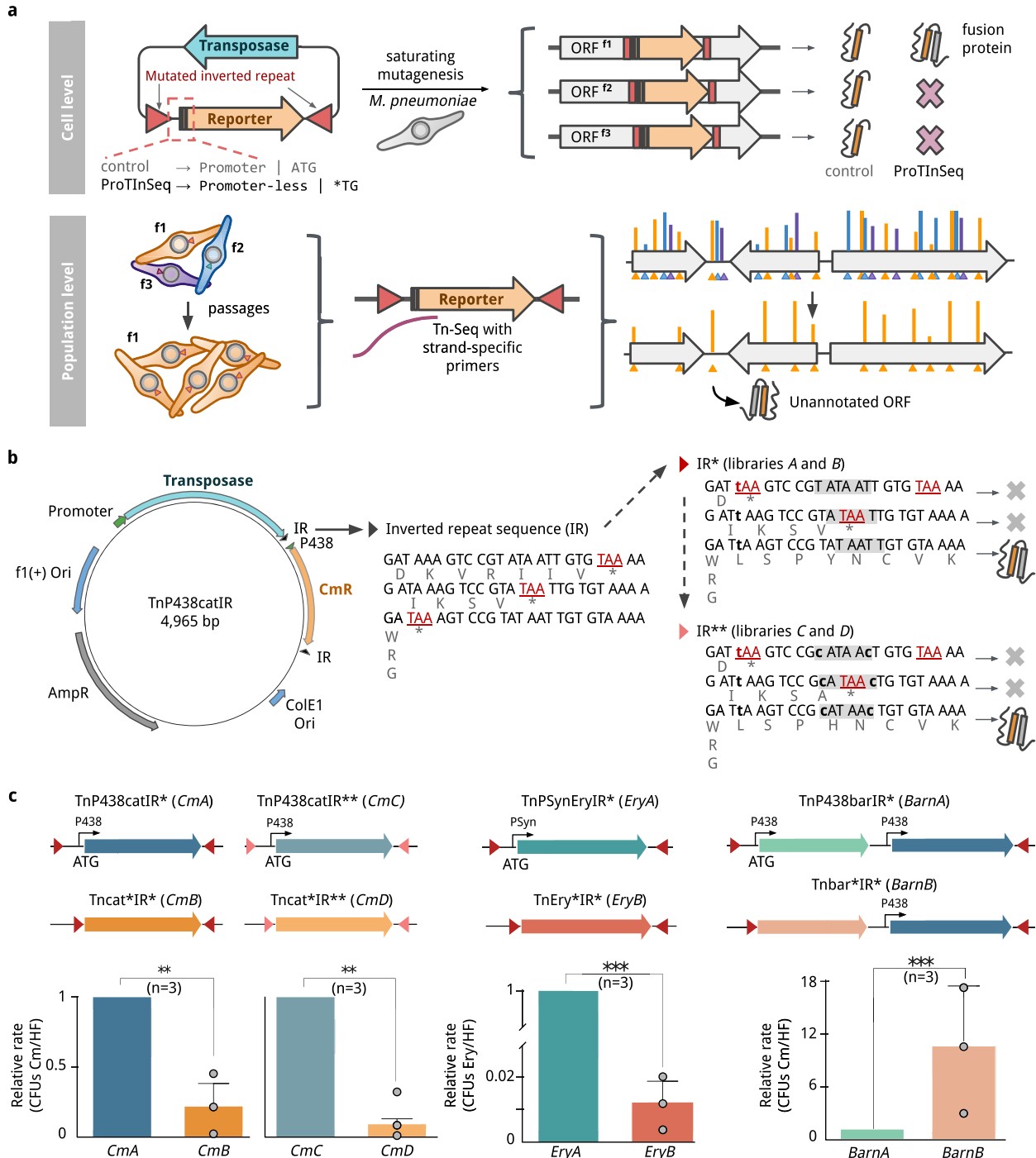

**Fig. 1 | Schematics of the ProTInSeq method and culture validation. a** At cell level (top), a reporter (orange) is expressed independently in a Tn-Seq protocol; in contrast, its initiation codon and the inverted repeats (IR) are mutated in Pro-TInSeq, so that the reporter is only expressed when inserted in-frame to an endogenous protein. At population level (bottom), individual transposition events occur in the population (orange cells, insertion in-frame; blue and purple, insertion in non-coding frames). Only cells expressing the reporter are viable when growing with an antibiotic. **b** The transposase (light blue) inserts a chloramphenicol resistance gene (orange). The original IR includes three stop codons (stop codons, red; translated sequence, gray), a mutation of A → t (first dashed line; IR*; red triangle) make the protein fusion to be in one of the three reading frames (libraries A and B). A second pair of T → c mutations to disrupt a Pribnow sequence (gray highlight) are introduced to create libraries C and D (IR**; light red triangle). **c** Schematics of the insert and transformation efficiencies of the libraries. Comparatives include 3 control and 3 mutated samples. First row shows control libraries with mutated IR* (red triangles) or IR** (light red), and the selection marker expressed under regulation of P438 or PSyn promoters. Second row shows the libraries without promoter and initiation codon. Last row, the histograms of ratios of the counted colony-forming units (CFUs) in the presence of the selection marker, normalized to numbers in Hayflick media. Bar plots heights represent the mean values with confidence interval as error bars. Statistical comparison by paired one-tailed T-test supports a significant decrease in relative transformation rate of *CmB*, *CmD*, and *EryB* with respect to *CmA* (P = 0.018), *CmC* (P = 0.013) and *EryA* (P = 5e-9). The negative-selection *BarnA* library has low transformation efficiency; once barnase is inactivated by mutation, the number of recovered transformants also increases (P = 0.05). Source data are provided as a Source Data file.

**Table 1 | Insertions found by ProTInSeq and validated by Sanger sequencing**

| Position in *MPN* genome [bp] | ORF inserted | Nr times sequenced | Function | Essentiality category |
|---|---|---|---|---|
| 469,307 | MPNs02 | 1 | Unannotated SEP | NE |
| 546,063 | MPN447 | 1 | Hmw1 (adhesion) | NE |
| 218,775 | MPN165 | 1 | Ribosomal protein L3 (rplC) | E |
| 751,237 | MPN624 | 3 | Ribosomal protein L28 (rmpB) | F |

## A sequencing approach to explore protein translation in bacteria

We performed DNA deep-sequencing of 39 *MPN* transposon libraries, including 2-3 biological replicas, that comprised the different selection reporters (*CmA, CmB, CmC, CmD, EryA, EryB, BarnA* and *BarnB*), combined in the case of chloramphenicol, with different concentrations in the cell culture (0.5, 1, 2, 5, 10 and 15 µg/ml) to define the best antibiotic concentration required to retrieve a significant in-frame selection of insertions, provide enough insertion coverage and be used to quantify protein expression. Transposon-inserted sites, insert orientation, in-frame insertions and their read count were identified by FASTQINS[31] looking for library-specific IRs (see Methods; Supplementary Data 2 and 3). For each sample, referred as *ReporterType-Concentration* (e.g. CmB15 corresponds to *CmB* grown with 15 µg/ml of chloramphenicol), we calculated the insertion *coverage* (i.e. percentage of insertion sites) and distinguished inserted positions in the genome by: i) *annotated* for in-frame codon positions of annotated ORFs, ii) *non-coding* for positions with no possible ORF and iii) *putative* for all in-frame, possible non-annotated ORFs of *MPN* (n = 29,424, see Methods). In this way, we evaluated the selection at the genome level by comparing the different groups (Supplementary Data 4).

No significant differences in the three categories explored were found in the controls *CmA, CmC* and *EryA* in terms of genome insertion *coverage*, supporting an homogenous rate of insertions along the genome (Fig. 2a and Supplementary Fig. 1). The maximum *coverage* was recovered at 15 µg/ml of *Cm* for *CmA* (1 insertion every ~3 bp, or 1 insertion every 2 bp when excluding E genes). In terms of genome *coverage*, at *Cm* concentrations of 0.5 or 1 µg/ml we observed no significant differences between *CmB* and *CmD* libraries and their respective controls *CmA* and *CmC* ($P > 0.05$; Supplementary Data 5). Increasing the *Cm* concentration ($\geq$2 µg/ml) led to significantly lower *coverages* for *CmB* with respect to *CmA* control, indicating antibiotic selection (Fig. 2a). For the *CmB* samples, we observed a significant relative increase toward in-frame insertions between *annotated* and *non-coding* positions in *coverage* for all the biological replicas of *CmB5, CmB10* and *CmB15* (Fig. 2a, Supplementary Data 4 and 5). We observed that insertion coverage in *non-coding* positions in libraries B and D plateau at 1 insertion every 100 bp for $\geq$5 µg/ml of chloramphenicol (Fig. 2a); thus, suggesting that this is the technical noise level. Although selection was already observed at 2 µg/ml of chloramphenicol, only samples with 5, 10, and 15 µg/ml were considered in further analyses. For the additional *CmD* (2 and 15 µg/ml) and *EryB* libraries (0.02 and 1.5 µg/ml), we observed the same enrichment patterns for *annotated* in-frame insertions compared to *non-coding* regions (Supplementary Fig. 1).

Then we inspected the NCBI annotated 690 protein-coding sequences of *MPN*, distinguishing them by their essentiality ($n_E$ = 299, F $n_F$ = 59, $n_{NE}$ = 332[29]). The results from the control libraries of *CmA5* (n = 3), *CmA10* (n = 2) and *CmA15* (n = 3) aligned with those from previous Tn-Seq experiments in *MPN*[29,31], with gene *coverages* following the expected E < F < NE distribution and no differences between the three ORFs in each protein-coding sequence (Fig. 2b). The data for the control

*CmA* library was consistent with previous studies showing preferential insertions in the 5% of each N- and C- terminal gene regions especially for E and F genes[31]. In the case of the *CmB* library, we only observed this phenomenon for the insertions in frame with the gene (Fig. 2b). When comparing the corresponding *CmA* samples to *CmB5, CmB10* and *CmB15*, we observed a significant reduction in gene *coverage* in off-frame positions of annotated ORFs of the *CmB* samples (one-tailed Mann-Whitney-U, $P < 0.05$ in every compared condition; Fig. 2b) while preserving comparable gene *coverages* between samples (Supplementary Fig. 2). Similar results were found for *EryB* (one-tailed Mann-Whitney-U, P = 0.001) while in the case of the *BarnB* samples (n = 3) we found a specular image with no insertions in-frame (Supplementary Fig. 3) and a significantly reduced number of in-frame insertions for annotated ORFs with respect to non-coding (Fig. 2c and Supplementary Data 5; one-tailed Mann-Whitney-U, $P = 0.011$).

For the *BarnB* library, we expected a genome *coverage* of ~18.6% (i.e. 2/3 of the insertions mapping to coding genes in *CmA15*) but instead we found a much lower *coverage*, of 0.5 ± 0.4% (three biological replicas; one-tailed Mann-Whitney-U, $P < 0.01$). This low genome *coverage* could be related to translational noise given the first gene in a transcript in *MPN* rarely depends on RBS (Supplementary Data 6, Shine-Dalgarno motif found in 26% of the genes[38]). To see if this was the case we selected another Mycoplasma species (*Mycoplasma agalactiae*) where 73% of the genes have RBS motifs at their 5′ UTR and needs a RBS for efficient expression of genes[38]. We transformed *M. agalactiae* with a *BarnB* vector in a modified version of Tn4001[38], which includes RBS motifs upstream to the transposase to efficiently transform this organism, and another upstream to the chloramphenicol acetyltransferase cassette used for selecting transformed cells and found downstream the barnase gene. As a control, we transformed both *Mycoplasma* species with the *BarnB* vector where the barnase RNAse catalytic activity is inactivated by the mutation H102A[45]. This experiment evidenced an approximate 10-fold significant increase in the ratio of viable colonies in *M. agalactiae* transformed with the *BarnB* transposon compared to the inactivated barnase control than in the case of *MPN* (Fig. 2d). This result supports the presence of significant translational noise in *MPN* due to the absence of RBS.

## Benchmark of ProTInSeq and comparative with other experimental methods in identifying translation of annotated proteins

We evaluated the transposon insertion frame preference to identify coding sequences, and especially, for the detection of SEPs. We analyzed the libraries where selection was significant (i.e. *CmB5, CmB10, CmB15, CmD15* and *EryB*). We also included one additional replica for *CmB5, CmB10*, and *CmB15*, in which we did three additional cell passages to remove any possible dead cells that could result in off-frame insertions. As a negative control, we used a list of strand-specific *non-coding* annotations derived from intergenic regions with low RNA expression profile ($\log_2$(CPM) < 2; negative set, n = 1700; Supplementary Data 6). We defined a method based on gene *coverage* that evaluates enrichment in the rate of insertion for in-frame ORF positions using the methodology applied to assign essentiality classes assuming that insertions in a gene follow a Poisson process conditioned by the gene length[29,31,46]. We defined two combined filtering steps to ensure we retrieve proteins with high confidence: i) a minimum number of insertions required, to ensure that negative control sequences are discarded; and ii) having a significant signal ($P < 0.05$) in at least two biological replicas or under different selection conditions (Supplementary Fig. 4, Supplementary Data 7). Under these criteria, we did not find false positives and reported a total of 518 coding sequences (CDS) in *MPN* (75% of the proteome, average per sample of 60.4%; Supplementary Data 8 and Supplementary Fig. 5), including 24 out of 27 annotated SEPs (88.9%; Fig. 3a and Supplementary Fig. 6).

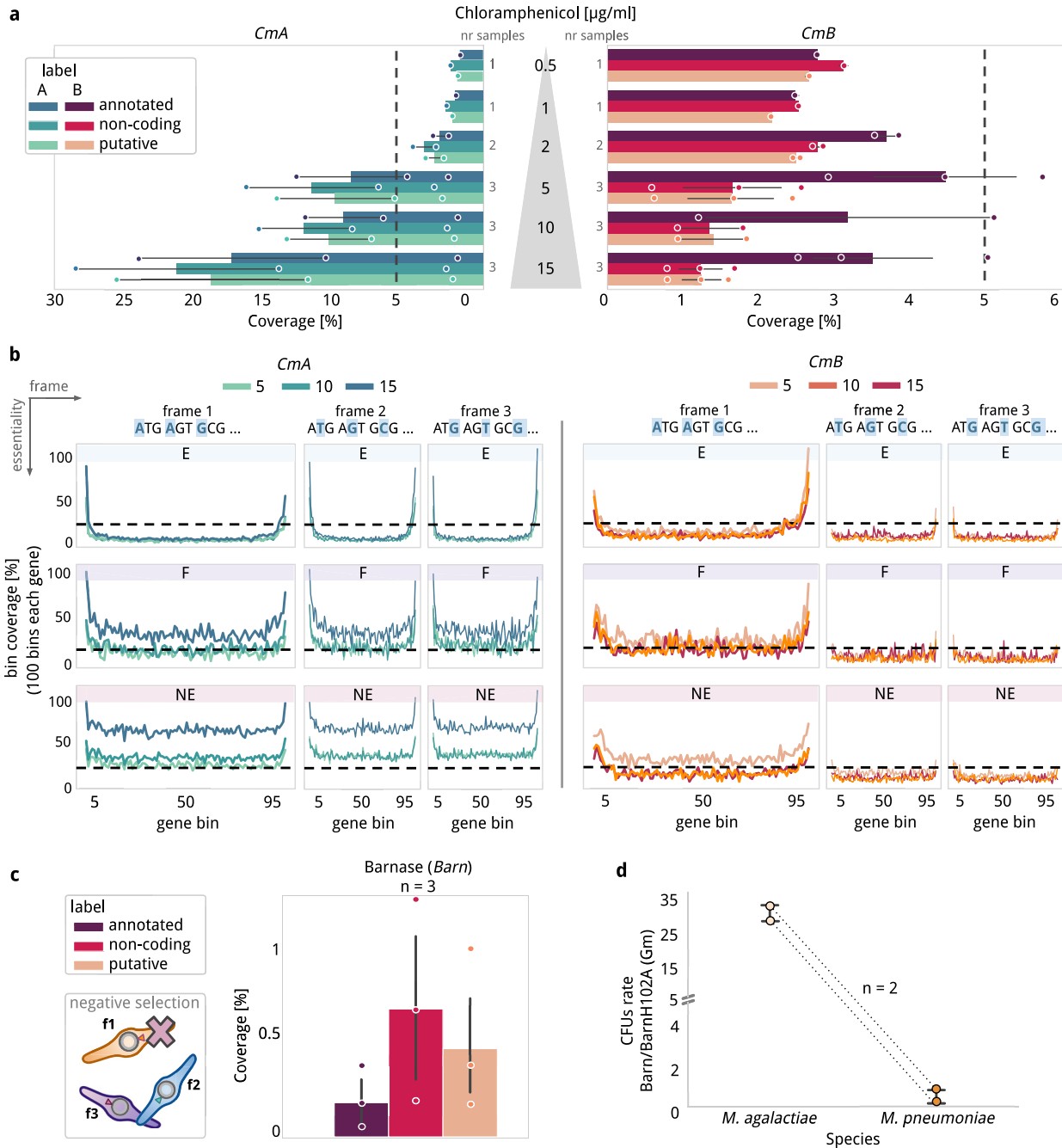

**Fig. 2 | Libraries comparative at genome and gene level. a** *Genome coverage* (X-axis), measured as number of insertions normalized by genome size (816,394) in *MPN*, compared along 6 chloramphenicol concentrations (Y-axis) between control expressing the resistance constitutively (*CmA*, blue to green) and the version that only expresses it in-frame with an endogenous gene (*CmB*, red to orange). Number of biological independent replicates are included in the base of the bar plots representing the mean values +/- confidence interval as error bars. *CmB* inserts in-frame to annotated genes at chloramphenicol concentrations higher than 2 μg/ml (one-tail Mann-Whitney-U *Cm5* (P = 0.001), *Cm10* (P = 0.04) and *Cm15* (P = 0.003)). **b** Metagene comparative of the *gene coverage* (Y-axis) calculated for genes in *MPN* binning them in 100 non-overlapping regions with the same size within the same gene (X-axis, from N-terminus to C-terminus). Values are min-max scaled for comparative purposes. *CmA* (left) *coverage* measured at concentrations of 5, 10 and

15 μg/ml of chloramphenicol (from light green to dark blue). No differences are observed between the three frames and the order of E < F < NE in *coverage* is respected. The same visualization is presented for *CmB* (right) from light orange to red for 5, 10, 15 μg/ml of chloramphenicol, respectively. It can be observed that frames 2 and 3 have almost no insertions while 'frame 1' corresponding to in-frame insertions resembles the *coverages* observed in the control. It can be noticed that there are preferential insertions at the C-terminus of E and F genes. **c** Same evaluation as panel A in with the barnase vector. Bar plots represent the mean values +/- confidence interval as error bars. In this case only cells expressing the insert in frames 2 or 3 are expected to survive (schema). **d** Ratio of CFUs obtained when comparing the number of CFUs obtained when transforming *M. agalactiae* or *MPN* with the vector having barnase (*BarnB*) and inactive barnase (BarnH102A) Both vectors do not have a starting codon. Source data are provided as a Source Data file.

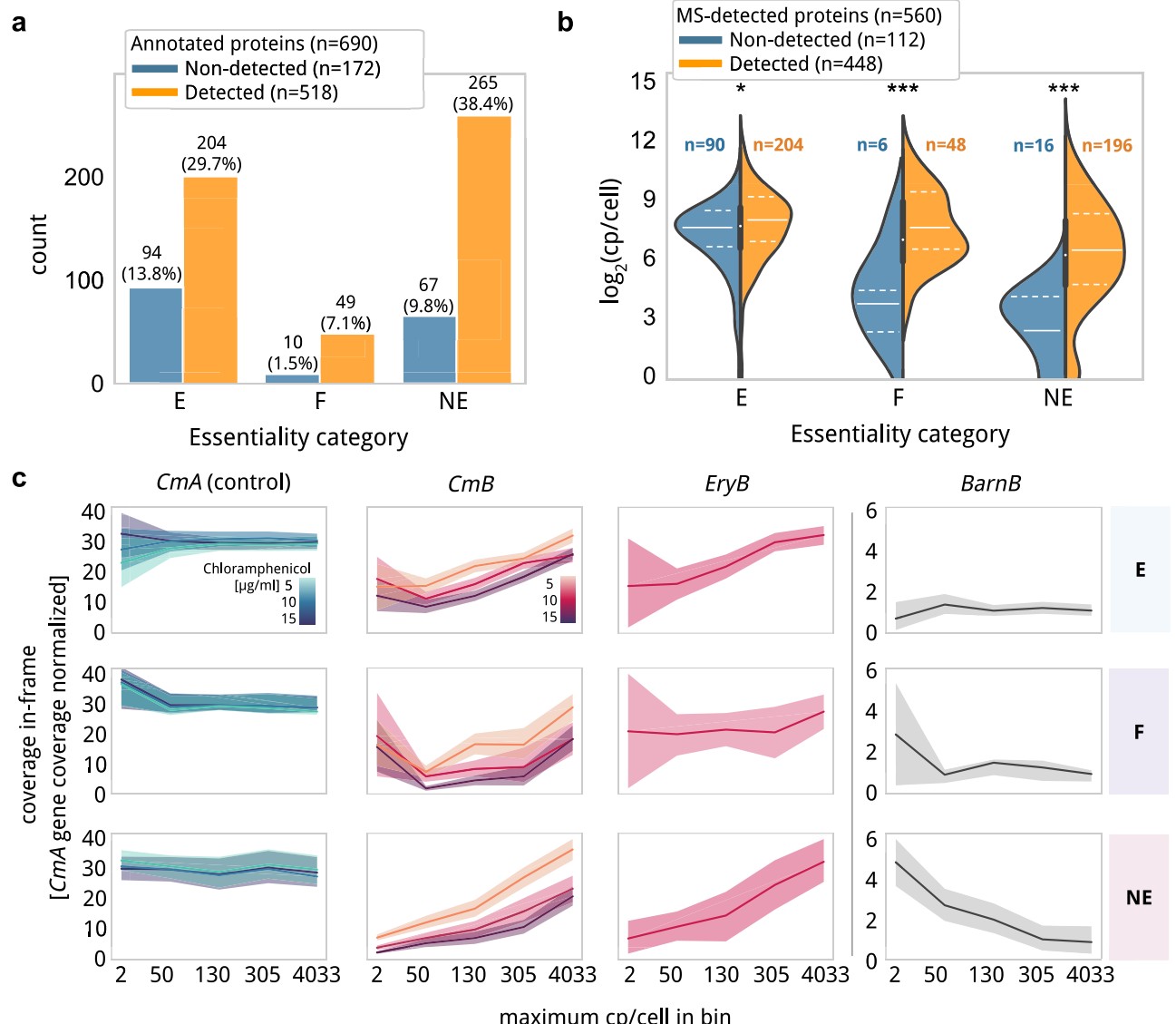

**Fig. 3 | Essentiality and protein levels in relation to ProTInSeq. a** Bar plot showing the number of annotated proteins in *MPN* detected as significant (orange) and non-detected (blue), divided by essentiality categories in *MPN*. Total counts and percentages over the total number of annotated genes in *MPN* ($n = 690$) are expressed above each bar. Note that most non-detected genes belong to the E category (E, essential; F, fitness; NE, nonessential). **b** Violinplot comparing the protein abundance of 560 proteins detected by mass spectroscopy with those detected by the ProTInSeq method (orange). Solid white line represents the median while the dashed lines are centered in the percentiles 25 and 75. Inner box plot represents the combined distributions. Group F and NE non-detected proteins are mostly low-expressed proteins (one-tailed Mann-Whitney-U; $P = 0.009$, 0.001, 0.002, for E, F and NE, respectively). For E-category genes, ProTInSeq non-detection is mainly due to the low number of insertions accumulated by these genes. **c** Relationship between protein abundance (X-axis) and the essentiality categories E, F and NE (rows) for the *CmA, CmB, eryB*, or *BarnB* library (columns). The X-axis represents 5 balanced bins (approximately 112 proteins per bin) and the maximum cp/cell found in the bin is used as reference labels. The Y-axis represents the *coverage* in-frame normalized by the *coverage* measured along the whole gene in the respective antibiotic concentration of the *CmA* control libraries (for random distribution a value of 33 is expected). Lines represent the average *coverage* for the genes in each abundance bin for a specific antibiotic concentration and the shadow represents the 95% confidence interval. When the shadows between two lines do not intersect (e.g. *CmB5* with *CmB10* and *CmB15* NE genes), the differences are considered significant. Source data are provided as a Source Data file.

In a pool of 116 MS samples, 538 proteins and 22 SEPs out of 27 annotated in this bacterium were detected[10]. Missing ones present either few UTPs, or represent fragmented genes (pseudogenes), or duplicated proteins[10]. Considering NCBI annotated protein-coding genes, 448 annotated proteins (including 21 SEPs) were found in the intersection between MS-detected and proteins identified with ProTInSeq. A total of 112 proteins and 1 SEP (*mpn169*, an E ribosomal protein of 87 aa) were exclusively detected by MS. These proteins are characterized for being either E ($n = 91$) and/or significantly low-expressed proteins (Fig. 3b). A total of 70 proteins (68 NE and 2 F) were

found with ProTInSeq and not by MS. Of these, 30 are hypothetical proteins, and 40 present diverse annotated functions in at least 2 closely related Mycoplasma species (Supplementary Data 7). Interestingly this group contained 24 out of the 28 proteins that could only be detected by MS when the Lon protease was knocked out, indicating that they are unstable and/or fastly degraded in the cell under normal conditions[47]. Considering that these proteins presented a gene *coverage* enrichment at the end of N′ and C′-termini regions compared to the control (one-tailed paired Mann-Whitney-U; $P < 0.001$ in each comparison with *CmB5*), it is possible that the fusion of the reporter

stabilizes and protects them against Lon targeting as the signal for degradation is in many cases located in the C′-termini of some of these genes (e.g. *FtsZ* and *FtsA*[47]; Supplementary Data 6). Finally, a total of 57 annotated CDS were not detected either by MS or by ProTInSeq. This included 11 pseudogenes; adhesins ($n = 18$) and hypothetical proteins ($n = 17$) containing repeated sequences, a factor that limits sequencing and MS approaches, as a fewer number of unique reads/peptides can be detected, respectively[10,31]. In addition, these 57 CDS presented low RNA expression values ($\log_2$(CPM) 2.6) compared to the average ($\log_2$ (CPM) = 7.5), indicating that they are probably not expressed under the experimental conditions.

In addition, we performed a comparative against Ribo-Seq data from *MPN* published in a recent study[48]. For this, we defined a protein as identified when the average ribosome coverage, or *RCV* (i.e. number of ribosomes normalized by bp length of an annotation, Supplementary Data 9) was larger than the background found in intergenic annotations between operons, rRNAs, tRNAs, and ncRNAs in *MPN* allowing for a 5% of error (see Methods). A similar number of annotated proteins than the ones reported by ProTInSeq ($n = 518$) were identified by Ribo-Seq ($n = 506$), with an overlap of 398 proteins identified by both Ribo-Seq and ProTInSeq. When adding MS data to this comparative, a total set of 375 proteins were retrieved by the three methods independently. These proteins presented significantly higher copies per cell measured by MS (average 328 copies per cell, while the general average is 210, Mann-Whitney-U test, $P = 2.09 \times 10^{-14}$; Supplementary Data 9). MS and Ribo-Seq identified 93 proteins not detected by ProTInSeq (all E and F genes). MS and ProTInSeq identified 73 proteins not detected by Ribo-Seq. Finally, of the NCBI annotated proteins 19 were found by MS, 15 by Ribo-Seq, and 47 by ProTInSeq (9 of them characterized as pseudogenes, or splitted genes, from out of a total of 35 of this type of genes in *MPN*) that were not identified by any other method. These 47 proteins presented significant lower RNA expression values (Mann-Whitney-U test, $P = 3.3 \times 10^{-17}$; Supplementary Data 9), highlighting the value of ProTInSeq as a complementary approach to Ribo-Seq and MS (details of the intersections found in Supplementary Fig. 6). Furthermore, we observed that a lower number of reads sequenced and recovered transposon insertions than ribosome footprints were required to achieve similar performance using ProTInSeq (Supplementary Fig. 7).

In conclusion, ProTInSeq compares similarly, in terms of detected annotated proteins in *MPN*, to MS and Ribo-Seq, and can be used as an orthogonal method to identify proteins overlooked by other experimental high-throughput methods.

### Factors affecting protein identification by ProTInSeq and its use for relative quantification of expressed proteins

We next explored the different biological factors that could play a role in the detection of proteins by our method by applying Principal Component Analysis (PCA). We looked at *coverage* in annotated genes and biological features including localization, RNA expression, protein abundance, membrane topology, essentiality (measured in non-mutated versions), and conservation. We found that the two primary components, accounting for 85% of the observed variability (54% and 31%, respectively), were primarily influenced by protein abundance and essentiality in the first component, and the presence of transmembrane segments in the second component (see Methods). As expected, due to the nature of this methodology, the E category comprised the most missed genes (13.8%, Fig. 3a). With respect to F and NE genes, we observed that the group of non-detected proteins by ProTInSeq presented lower protein abundance levels (one-tailed Mann-Whitney-U, $P < 0.05$ for each comparison) with respect to the detected (Fig. 3b; Supplementary Data 7). This suggests that insertions in genes with low expression are not favored, as the reporter, when in-frame, fails to attain the necessary levels for resistance. In a conventional Tn-Seq experiment, we would expect to see higher *coverages* for

NE with respect to F proteins. However, as F proteins are expressed on average at higher levels than NE proteins (F = 304.9 and NE = 212.3, average copies/cell; one-tailed Mann-Whitney-U test, $P = 0.003$), this gene category present comparable *coverages* to NE genes in the *CmB10* and *CmB15* samples (Fig. 2b). In agreement with this, we observed in the *CmB5* samples higher *coverage* values for proteins with lower abundances compared to *CmB10* and *CmB15* samples (Fig. 3c, $P = 0.001$ in every condition). In the *BarnB* library, E and F genes presented residual *coverage*. For NE genes, those with higher protein copy numbers per cell presented lower *coverages* compared to those with low copy numbers (one-tailed Mann-Whitney-U test; $P = 0.03$), indicating that *Barn* can be inserted in genes with very little expression (Fig. 3c). We evaluated this effect by exploring the relation of *coverage* in-frame, normalized by essentiality, with protein abundances (explored in 5 bins; Fig. 3c). This revealed that while gene *coverage* in the control *CmA* sample remained comparable along with the protein copies per cell groups, *CmB* and *EryB* gene *coverage* increased in general with protein abundance; this indicated that, in addition to essentiality, the library selection also depends on the levels of protein expression (Supplementary Fig. 8).

The capability for relative quantification of protein abundance by ProTInSeq was evaluated in comparison with Ribo-Seq (see Methods; Supplementary Data 9) by measuring the Spearman correlation coefficient with the protein copies per cell of 560 *MPN* proteins measured from 116 MS experiments[10]. When taking all these proteins together, ProTInSeq showed a limited correlation with protein abundances (R = 0.38), smaller than that obtained when using Ribo-Seq *RCV* (R = 0.65) and those same values normalized by RNA expression (R = 0.47). However, when assessing the correlation values separated by essentiality categories, we observed that these results were mainly lowered by E genes (ProTInSeq R = 0.28, Ribo-Seq R = 0.49). ProTInSeq showed comparable results to Ribo-Seq when exploring F genes (ProTInSeq R = 0.70, Ribo-Seq R = 0.64) and NE (ProTInSeq R = 0.67, Ribo-Seq R = 0.64); highlighting the complementarity of these sequencing methodologies (Supplementary Fig. 9).

### ProTInSeq insertion profiles represented transmembrane segments for NE proteins

We observed proteins with one or more transmembrane segments presenting differences of gene insertion *coverage* between the positive control and selection libraries. *MPN* has 41 annotated lipoproteins (35 NE and 6 E) and they all have low *coverage* in *CmB* samples (Supplementary Data 7). Lipoproteins are synthesized with an N-terminal region encompassing a transmembrane segment cleaved by a peptidase when acylating a cysteine residue[49] and being fully exposed to the outside of the cell[49]. Thus, we would only expect insertions under antibiotic selection at the N′-terminus, or in the case of barnase in the external region (if 100% of the protein goes outside of the cell). While the control *CmA* library presented a homogenous *coverage* along the NE lipoprotein genes, we observed only insertions in the N-terminus in the *CmB* library (Fig. 4a and b). Another specific effect was observed for other proteins with more than one transmembrane segment. When measuring the in-frame *coverage* in predicted cytoplasmic segments of 66 NE genes that encode for proteins with at least 2 predicted transmembrane segments (according to TMHMM) normalized by their total in-frame *coverage* in *CmB15*, an average of 81% ± 17% of the total insertions was found in the predicted cytoplasmic segments of the gene. Opposite to this, barnase insertion enrichments were found in the predicted extracellular segments of the genes offering complementary validating information on membrane topology. Applying a Change Point Detection (CPD) algorithm to detect significant deviations in continuous data[31], we could detect the predicted transmembrane segments (Fig. 4c). In some cases, such as for the *mpn359* gene, our results contradicted the predictions made by TMHMM (60% of in-frame insertions were in outer-segment coding regions; Fig. 4c).

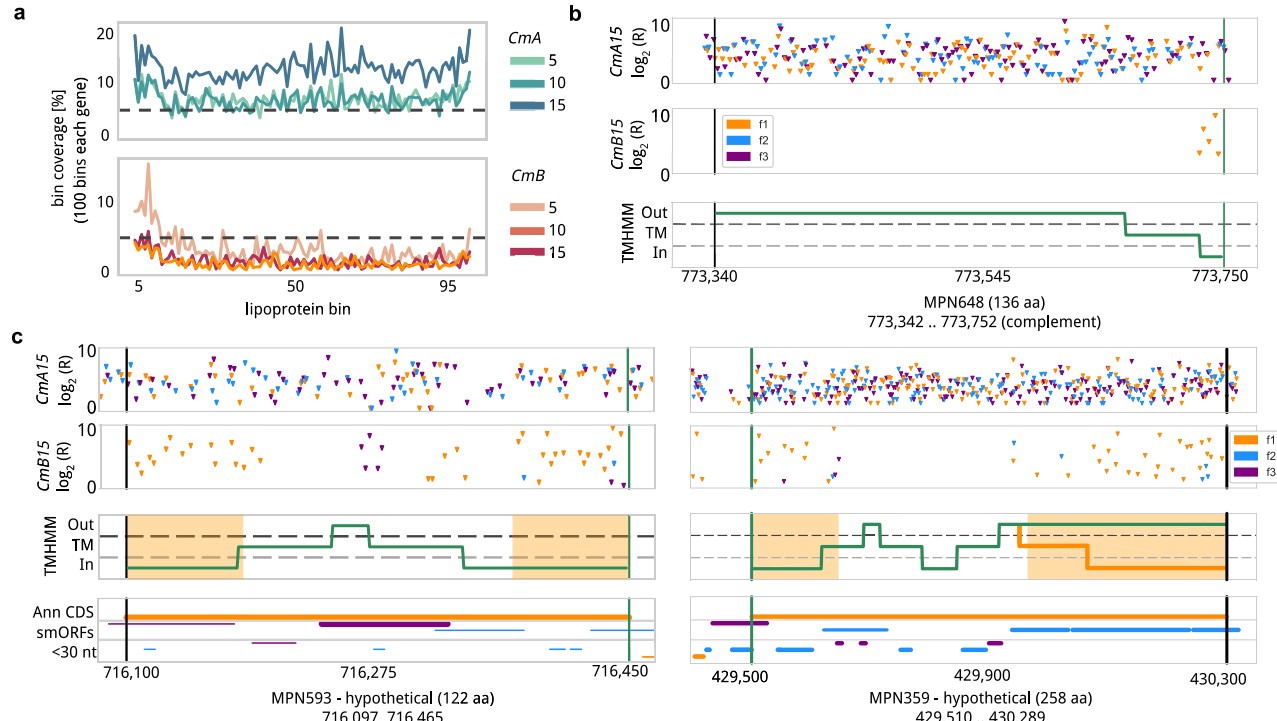

**Fig. 4 | Transmembrane topology exploration using ProTInSeq. a** Metagene representation of the *coverage* in in-frame positions of 35 NE lipoproteins comparing libraries *CmA5* and *CmB5*. While insertions are homogeneously distributed in the *CmA* control, insertions in the *CmB* are only recovered in the N'-terminus. **b** Top two plots: Insertion profile in the *mpn648* encoding for a lipoprotein (X-axis represents the position in the chromosome, green and black vertical lines represent start and stop codons, respectively). Y-axis represents the log₂ of average *read counts* (R) of positions found inserted in at least two replicates (f1,2,3 are frames 1, 2 and 3) The top two plots are for *CmA15* and *CmB15* samples, respectively. One can see only in-frame insertions at the N-terminus of the protein. The lower plot presents transmembrane segments predicted with TMHMM. Regions located in the cytoplasm (In) are located below the gray-dashed horizontal line; Solvent-exposed

regions (Out), are above the black-dashed horizontal line. Transmembrane segments (TM) are in between the two dashed lines. The region with insertions corresponds to the predicted cytoplasmic sequence. **c** Same representation for the *mpn593* (left) and *mpn359* (right) genes encoding for membrane proteins with more than one transmembrane segment. While in the case of mpn593 an agreement between the predicted and experimentally detected cytoplasmic and external regions is observed, this is not the case for the predicted second outer segment of *mpn359* (in green prediction by TMHMM, orange line for the prediction by SPLIT). In the case of *mpn593* there seems to be a smORF in the external predicted region (purple insertions in *CmB15*). If an ORF is significant based on the insertions in the frame, the line is broader. start stop is delimited by green and black horizontal lines. Source data are provided as a Source Data file.

This could be a consequence of a wrong prediction by the software used. In fact, using the SPLIT server[50], we found a putative fourth transmembrane helix with a weak prediction, which, if true, will indicate that the C-terminal region of the protein is internal as supported by our data (Supplementary Fig. 10).

## Identification of unannotated proteins and SEPs using ProTInSeq

Using the same criteria defined above to decide if an ORF encodes for a polypeptide, we identified 158 non-annotated ORFs (Supplementary Data 10). This list included 5 ORFs (>300 nt), of which 3 can also been detected by MS; these are predicted to encode proteins of 104 aa (*mpneu10249*, 3 UTPs), 193 aa (*mpneu25274*; 5 UTPs, GTG start codon; predicted dihydroxyacetone kinase subunit L by BLASTP; Supplementary Fig. 11) and 252 aa (*mpneu06085*; 8 UTPs, also the largest ORF in this group; a predicted lipoprotein). The detected 153 SEPs ranged between 9 aa to 95 aa in size (40 ± 20 aa, median = 38). Fifty-four of these had already been reported in a list of 118 computationally predicted coding unannotated SEPs[10]. Three of these 54, were previously validated by C₁₃-labeled peptides: *mpneu00732* (MPN155a; 90 aa; 2 UTPs; YlxR, RNA binding protein), *mpneu14551* (MPN655b; 82 aa), and *mpneu14957* (MPN672a; 57 aa)[10]. Only 7 of the 153 SEP candidates presented low RNA expression profiles in the range of what we considered as a negative control (log₂(reads)<2), while the remaining 146

showed an average expression of 8.7 ± 2.5 log₂ (reads), which is higher than that of annotated genes encoding for proteins (i.e. 6.7 ± 2.4 log₂ (reads)).

*MPN* does not rely on RBS present in the 5′UTR of the first gene of an operon to initiate translation but could contain those sequences at the 5′ of genes within an operon. In agreement with this, we found 34 smORFs (22%) presenting a RBS sequence and being expressed within known operons. About half (46%, *n* = 71) of the putative SEPs were in transcribed intergenic regions (e.g. *mpneu14402* in Fig. 5a). We found 23 smORFs (15%) overlapping with annotated ncRNAs in *MPN*[51,52], including for instance *mpneu12044* (69 aa), *mpneu02279* (14 aa) and *mpneu07215* (10 aa) which overlap with *ncMPN037* (Fig. 5b). The fact that SEPs would be encoded in the same transcript suggests they are expressed in the same operon and might be coregulated. Additionally, we found 10 cases (6%) in which a identified SEP would be expressed no further than 10 bp from an annotated gene start codon suggesting they could have a regulatory role in the translation of upstream genes by hiding/exposing genetic signals when being translated[53] (e.g. *mpneu07215*; Fig. 5b). Finally, 38 smORFs (24.5%) overlap with larger annotated genes (Fig. 5c).

We then compared ProTInSeq as a method for validation of the expression of SEPs with Ribo-Seq, the state-of-the-art sequencing approach for SEPs identification using the criteria applied for annotated proteins. Ribo-Seq reported evidence of translation for 95 SEPs.

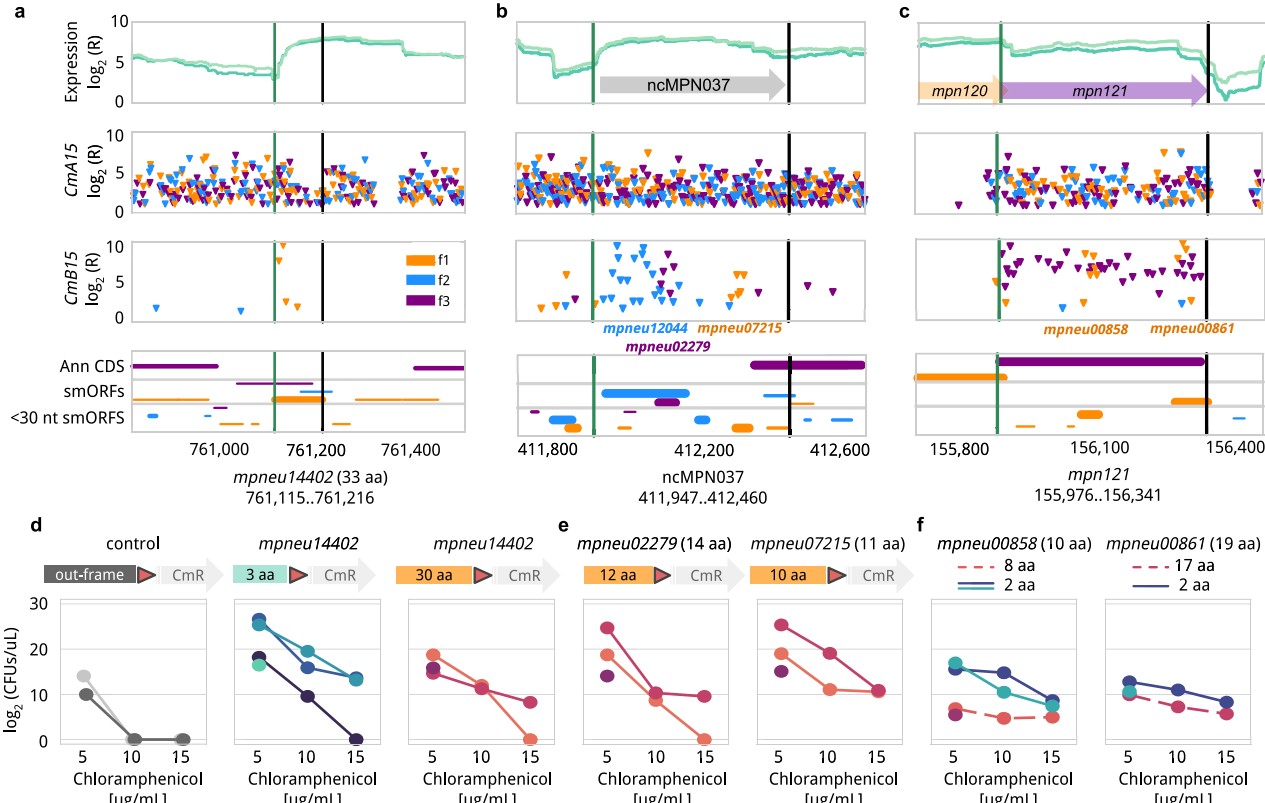

**Fig. 5 | Examples of profiles of smORFs detected to translate to SEPs with this approach.** In **a–c** we show from top to bottom the RNA sequencing profile as log2(reads) along the genome (X-axis), then the insertion profiles (log2(reads)) obtained by Tn-Seq from the *CmA15* and *CmB15* samples. Each inverted triangle represents an insertion found in at least two replicates. The colors represent the 3 possible frames of the whole region: frame 1 (orange), 2 (blue) and 3 (purple). The final row shows the ORFs found, with the same frame color code as in the upper plots, distinguished by annotated ('Ann CDS'), smORFs between 30 and 300 bp and very small ORFs (15–30 bp). If an ORF is significant based on the insertions in the frame, the line is broader. start stop is delimited by green and black horizontal lines. **a,** *mpneu14402* in orange; predicted as a SEP with an ATP-binding domain by BLAST.

**b**, profile of ncMPN037 which has 3 overlapping SEPs: *mpneu12044* (69 aa), *mpneu02279* (14 aa) and *mpneu07215* (11 aa) **c**, example of smORFs (*mpneu00858* (10 aa) and *mpneu00861* (19 aa)) overlapping with a coding gene, *mpn121*. **d–f**, validation of new smORFs as coding genes. X-axis shows the increasing concentrations of the antibiotic used. Y-axis, log2 of the number of colonies forming units (CFUs) per μL. The left arrows show the length of the smORFs fused to the CmR gene (**d** and **e**), or we show the length in numbers (**f**). The different lines in the plots indicate biological replicas and/or fusion positions. As control (**d**, first panel), we repeated the experiment with *mpneu14402* but leaving the resistance gene out of phase by 1 (light gray) or 2 bases (dark gray). Source data are provided as a Source Data file.

Due to the lack of in-frame footprint selection in *MPN* (Supplementary Fig. 12 and Supplementary Data 10), smORFs overlapping larger genes were not possible to assess with this approach. None of the SEPs identified with at least 2 UTPs, nor those validated by C13-labeled peptides, MS[10], and culturing (see section below), were detected in the Ribo-Seq analysis. Of the 95 Ribo-Seq SEPs, 13 overlap with the ProTInSeq hits, and 14 were predicted by RanSEPs[10]. When combining SEPs detected by the approaches considered (Supplementary Data 10, see Methods), we obtained a set of 302 unique SEPs. Out of these, 159 were supported by at least 2 independent methods. ProTInSeq was the method presenting the highest number of uniquely identified SEPs (*n* = 65, 54 predicted by RanSEPs), followed by Ribo-Seq (*n* = 60). A summary of the intersections between methods can be found in Supplementary Fig. 13.

As a summary, ProTInSeq identifies 22 of the 27 annotated NCBI SEPs (the ones missing are E genes; Ribo-Seq identifies 25) and identifies 3 of the 11 unannotated identified SEPs by MS[10] (none of these detected by Ribo-Seq). Thus, ProTInSeq is shown to be a valid orthogonal method to Ribo-Seq and MS that, in combination to these and computational approaches, provide support for 302 unannotated SEPs expressed in *MPN* representing an increase of 43.7% in coding potential over the initially annotated 690 genes in this bacterium.

## Function exploration and validation of unannotated SEPs by computational and experimental analyses

We explored the set of non-annotated 302 SEPs identified by all methods in *MPN* by different computational prediction approaches (Supplementary Data 11, see Methods). Conservation by BLASTP with the translated ORFomes from 109 bacterial species[10] returned 178 hits found to be conserved in at least two bacterial species, including 14 SEPs reported as hypothetical in other bacterial species, and 6 with a function annotated in NCBI in a different species. When excluding closely related species from the Mollicutes class (mycoplasmas and ureaplasmas), a total of 41 SEPs presented a hit in an evolutionary-distant species. Furthermore, we applied several protein function methods based on structure, protein domains, and orthology allowing us to provide insights on potential functions for 116 SEPs (Supplementary Fig. 13). Specifically, as reported in previous studies[10,19,20], a significant proportion of these SEPs (*n* = 81; 26%) presented sequence features corresponding to antimicrobial peptides by using the predictor AMPred[54] or signal peptide features[55] (*n* = 25, 8%). PfamScan[56] found conserved motifs in 12 SEPs and structure-based function prediction using DeepFRI[57] highlighted 10 SEPs with diverse predicted molecular functions such as ion, nucleic acid, protein or organic compound binding, and SEPs that could act as structural constituents

of the ribosome. Homologous sequence search in the UniProt database using PANNZER2[58] assigned functions to 10 SEPs and orthologous search with EggNOG[59] reported 9 functions for these unannotated SEPs[59] (Supplementary Fig. 14, Supplementary Data 11). As a summary of the 302 putative unannotated SEPs we could find some functional evidence supporting them for 116. We observed a higher number of SEPs with a predicted functional property for those identified by ProTInSeq ($n = 50$; 32% of 153; 16% of 302) than the SEPs identified by Ribo-Seq ($n = 24$; 25% of 95; 7% of 302).

Finally, to experimentally validate our newly identified potential SEPs we selected five of them: *mpneu14402*, *mpneu02279*, *mpneu07215*, *mpneu00858* and *mpneu00861*. These examples cover the three possible genomic contexts for SEPs, including independent expression, upstream regulator, and overlapping larger genes. We then cloned them using their closest promoter region in the genome supported by RNA-Seq and sequence motif prediction[60], its 5'-untranslated region, its start codon and part of the smORF followed by the mutated IR and *Cm* resistance gene in frame (Fig. 5d–f). For the control experiment we left the predicted SEP out of frame with the *Cm* resistance (Fig. 5d). We grew the corresponding clones under different Cm concentrations, plated the cells and the colony-forming units determined. In three cases (*mpneu14402*, *mpneu02279*, *mpneu07215*; Fig. 5d, e) we found a significantly higher number of colonies at 10 µg/ml Cm than in the out of frame control. Remarkably, the differences in read count observed in the insertions mapped in *mpneu14402* well-corresponded to the difference in number of colonies obtained showing a higher number when inserted at the N-terminus compared to the C-terminus (Fig. 5d). In the other two (*mpneu00858* and *mpneu00861*; Fig. 5f) the results were not significantly different from those of the control. Both *mpneu00858* and *mpneu00861* have a stretch of hydrophobic residues at their N-termini (Supplementary Fig. 15). Considering the Lon protease recognizes exposed hydrophobic protein tails[47], this could result in fast degradation of these SEPs that are identified only when the Cm resistance is inserted near the N-terminus. In agreement with this hypothesis, constructs where we inserted the *Cm* gene after the first two aa of these genes resulted in a similar behavior as the other three tested (Fig. 5f). This, together with the fact that unstable proteins only detected by MS when Lon protease is downregulated are detected by our method, suggests an inherent regulatory mechanism in *MPN*. This mechanism based on Lon quality control will result in fast degradation of produced peptides resulting from translational noise.

Overall, these results show that ProTInSeq identifies unannotated SEPs as well or even better than Ribo-Seq, with a remarkable percentage of them presenting functional potential predicted by different computational tools including high-value targets such as antimicrobials.

## Discussion

Identification of all protein-encoding ORFs in an organism is not a simple task and it is paramount in the understanding of the biological functions it can perform. Here we have developed a method that combines random transposon insertion with ultra-deep sequencing, ProTInSeq, to identify coding genes, especially those encoding for SEPs. This method uses a selection marker adjacent to the mutated IR of the mini-transposon that when in frame with a coding gene allows cell survival or, when introducing barnase, kills the cell. Insertions are shown to occur preferentially in in-frame positions for the positive selection (antibiotic resistance libraries) and the opposite for the negative selection barnase library. Using common approaches applied in the field of genome essentiality, we show insertion enrichments for ORFs and smORFs validated to encode for SEPs. In total, more than 7,000 translation signals could be recovered in *MPN* considering samples independently. To keep a conservative approach that considers the random nature of the original method, the possibility of

sequencing artifacts, and a small number of double transformations (~6% assuming a Poisson insertion process and the amount of plasmid DNA used), we provide a stringent criteria to identify translation events. We recover with ProTInSeq 75.2% of *MPN* proteome (83% of the proteins identified in 116 MS samples analyzed), including 66% of its annotated SEPs, 153 unannotated SEPs, 5 unannotated CDS, and 24 targets of Lon protease, which cannot be detected by MS as they are rapidly degraded. Within the group of 158 unannotated identified proteins, we observe a wide variety of sizes, from a SEP of 9 aa to a gene coding for a 252 aa protein. Out of the 153 unannotated SEPs, 54 were predicted by our published computational approach[10]. This difference could be explained since computational prediction relies strongly on conserved features, generally associated with E genes. The majority of the unannotated SEPs identified here had an associated transcript expressed at significant levels and found as smORFs in other bacterial genomes. Some of these unannotated SEPs were intergenic, while others overlapped with coding genes and ncRNAs, and with potential roles as upstream regulatory smORFs, not described in *MPN* but found in other organisms[53].

No distinction can be made at the moment between SEPs acting as functional proteins or those that are the result of translational noise, similarly to the presence of transcriptional noise in bacteria[35]. In *MPN* there is no need for a Shine-Dalgarno motif at the 5' of operons for the translation of the first gene to happen[38]. This could increase translational noise since the first ATG of the RNA will initiate translation. In fact, we see a significant degree of translation levels all over the genome of this bacterium supported by the high number of positive marker insertions and the low insertion coverage when using the barnase marker. This is important since the low expression of ORFs paves the way for the evolution of functions that, when needed, could be selected to increase their expression[23]. In *MPN*, smORFs could act both as a reservoir of new protein functions but also contribute to its slow growth compared to other Mycoplasmas with RBS dependence (e.g. *M. agalactiae* divides every 2–3 hours and *MPN* every 8 hours) due to the demands of managing transcriptional and translational noise. Alternatively, some translational noise could in fact help *MPN* in mitigating its significant transcriptional noise[35], as the translation of smORFs can provide more precise protein production of downstream genes[24]. Furthermore, Ribo-Seq and MS technologies have also reported translational signals that do not correspond to known proteins[61,62]. However, these cases are difficult to replicate between both approaches, either as a consequence of the short half-life and low abundance of the products translated[63], or because they are actually technical artifacts. Thus, applying ProTInSeq can help to distinguish actual translational noise from technical noise, even more considering that fusion of the selection marker could stabilize elusive proteins as shown with Lon protease targets here. Despite these observations, the high conservation of some of the identified SEPs and specific features associated with secretion or as antimicrobial peptides, indicates that certain SEPs cover specific functions in *MPN*.

As the expression of the resistance or anti-selection reporter is dependent on the expression of the protein to which it is fused, we could roughly estimate protein abundance by comparing insertions found in the controls and those in the selection libraries. We showed that, within essentiality categories, those genes with higher *coverages* are more abundant than those with lower *coverage*. This is exemplified by the fact that F genes retain similar *coverages* to those found in NE genes, as F proteins are present on average at higher copies per cell than NE proteins in *M. pneumoniae*. With the barnase library, we detected insertion in-frame in NE genes but only when they are very lowly expressed (<2 cps/cell). Thus, the analysis of the libraries described hereunder under different selection conditions could help in determining which proteins are being expressed and at which relative quantities. However, we should have a word of caution since we identified 71 F and NE genes that are not detected in free-label MS

searches. For example, we can detect in-frame insertions at the N-terminal region of a pseudogene with an internal stop codon not detected by MS but no insertions after the stop codon. Thus, in some cases, protein abundance determined by ProTInSeq could be affected by protein stabilization due to the fusion of the antibiotic selection marker. This can be exemplified by looking at FtsA and FtsZ genes that have degrons at their C-terminal, which result in low protein levels despite its high mRNA abundance[47]. In this case, we see preferential insertions at its C-terminus, and they can be identified at a high concentration of chloramphenicol despite the low expression levels in the WT strain. In terms of quantifying protein abundance, ProTInSeq retrieved comparable results to Ribo-Seq for F and NE genes but the correlation between protein copies per cell and ribosome footprints was low compared to studies in other systems. This last observation might be a consequence of the limited in-frame preference of ribosome profiling in *MPN* that would require alternative protocols, such as treatments to stall initiation complexes[64]. When exploring transmembrane and membrane-associated proteins, we observed insertion enrichments in the predicted cytoplasmic segments of these proteins compared to transmembrane and exposed segments. This happens because the fusions in the outer segment might expose to the medium the resistance enzyme and, consequently, the cell will die in the presence of antibiotics. This is well seen in the case of NE lipoproteins, which accumulate insertions only in their N'-termini regions, which is their only cytoplasmic segment. In the case of NE proteins with two or more transmembrane segments, we observed that the in-frame insertions correspond to cytoplasmic segments predicted by TMHMM. Taking advantage of this, we show we could use ProTInSeq to predict the topology of F and NE membrane proteins. Interestingly, using a negative marker, a specular image was observed, with insertions of barnase only retrieved for proteins not found expressed by MS or in protein segments exposed outside of the cell. Although the main purpose of the presented method is not quantifying protein abundance or predicting membrane topology, these are features that can be explored with the same experimental setup.

Altogether, ProTInSeq supports and complements proteomic information using ultra-deep sequencing samples with positive and negative selection reporters. It also allows the identification of new ORFs and smORFs in bacterial genomes, their relative quantification, and the determination of membrane topology features. The main limitation of this methodology is that it cannot identify essential genes (E) that do not have insertions at its extreme N and C-terminus positions. Furthermore, it needs to be acknowledged that, in contrast to alternative methods, this approach scales with the size of the target genome and efficiency of transformation of available transposons in the organism. Nevertheless, considering the diverse applications of Tn-Seq, such as conducting differential essentiality studies by culturing under different conditions, this technique has the capability to highlight condition-specific expression patterns of SEPs, offering insights into their functional roles that may remain elusive through alternative high-throughput methodologies. Finally, as this technique can, in principle, be applied to any genome, we envision ProTInSeq as an affordable asset in the experimental identification of SEPs orthogonal to methods such as Ribo-Seq and MS. Finally, prediction of protein function is still challenging, even more so in SEPs where there are less studied cases. Notwithstanding, ProTInSeq retrieved a larger number of SEPs with predicted function (32%) compared to Ribo-Seq (25%), indicating that this approach can highlight high-value SEPs targets for further specific inspection.

Ultimately, by combination of experimental and computational approaches, we provide support for 302 unannotated SEPs expressed in *MPN*, representing a 43.7% increase in coding potential compared to the initially annotated proteins in this bacterium. Out of these, 116 SEPs are predicted to have function by different computational estimators, antimicrobials being the top functional category. In conclusion, ProTInSeq can be used as an orthogonal method to identify expressed SEPs and elusive proteins at the same time it provides insights in proteome characteristics using a flexible and cost-effective DNA-sequencing approach.

## Methods
### Molecular cloning
To define the three libraries used in the current study (chloramphenicol, erythromycin and barnase libraries), we generated nine different constructs derived from the vector pMTnCat_BDPr[43]; a Tn4001-like mini-transposon vector (Supplementary Data 1). They were obtained by using the Gibson assembly (New England Biolabs) of three different fragments, following the instructions of the manufacturer. The purification of PCR products and digested fragments from agarose gels were achieved using the QIAquick Gel Extraction Kit (Quiagen). Plasmid DNA was obtained by using the QIAprep Spin Miniprep Kit (Quiagen). For the *Cm* and *Barn* libraries, the P$_{438}$ promoter was used[43], while for *Ery* we used the Psyn promoter[38]. To test the translational noise, we generated by Gibson 4 different transposons based on an universal transposon carrying the gentamicin resistance gene. In the first construct, we mutated the IR. In the second, we mutated the IR and we added the cassette for the fusion with the *Cm* mutated. In the third, we mutated the IR and we added the cassette for the fusion with the barnase, and in the fourth one we mutated the IR and we added the cassette for the fusion with the inactivated barnase. To validate the translated SEPs we designed transposon vectors replicating the fusion between the coding sequence and the mutated chloramphenicol resistance gene. Genomic DNA of *MPN* M129 was isolated with the Illustrabacteria genomic Kit (GE) and we amplified each smORF tested by PCR taking their regulatory regions (promoter, 5'UTR, and start codon) and a variable number of bases including the expected fusion and, as control, 1 or 2 nucleobases to disrupt it. Each PCR fragment was cloned by Gibson into a Tn4001 transposon (carrying the tetracycline resistance gene) with the mutated IR and the mutated *CmR*. For more detailed information about the designs and strategies of assembly see Supplementary Methods.

### Bacterial strains and growth conditions
*Escherichia coli* strain Top10 (Thermo Fisher) cells were grown at 37 °C in 2YT broth or LB agar plates containing 75 μg/ml ampicillin when needed. The *MPN* M129 strain was grown in 75 cm2 tissue culture flasks with 25 ml of modified Hayflick medium (HF) at 37 °C and was transformed as previously described[65]. To select *MPN* transformant cells, plates were supplemented with 20 μg/ml chloramphenicol or 0.02 μg/ml of erythromycin. Transformed cells were also grown in liquid cultures and testing different concentrations of antibiotics. First, *MPN* was grown in a 96-well plate format with 200 μl of HF and 5 μl of transformed cells. For chloramphenicol the tested concentrations were 0, 0.5, 1, 2, 5, 10, 15 and 20 μg/ml. In the case of erythromycin, the tested concentrations were 0,002 and 0.02 μg/ml. Concentrations of 0.5, 1, 2, 5, 10, 15 μg/ml for chloramphenicol libraries and 0.02 μg/ml for erythromycin libraries were selected for sequencing. To study the proteome of *MPN*, transformed cells were grown in T75 flasks with different antibiotic concentrations (0.5, 1, 2 and 15 μg/ml of chloramphenicol) to cover from low to highly expressed proteins. After 24 h cells were passed to a T300 cm$^2$ flask. Cultures of cells grown with 0.5, 1, 2 and 15 μg/ml of antibiotic were confluent after 48 h and cultures of cells grown in 15 μg/ml required three additional days.

### Transformation of *M. pneumoniae*
To generate the libraries, transformations of *MPN* were performed by electroporation as previously described[66] but with a slightly modified protocol. Briefly, cells grown in two T75 cm$^2$ flasks were recovered in a 2 ml electroporation buffer and 80 μl of cells were electroporated with 2 pmol of different vectors. After electroporation, cells were

resuspended in a final volume of 1 ml by adding 900 μl of HF. The 2 transformations of each vector were pooled (total volume of 2 ml). Five hundred μl of cells were cultured in 20 ml of medium in a T75 flask with different concentrations of chloramphenicol (0.5, 1, 2 and 15 μg/ml) for 4 days at 37 °C in 5% $CO_2$. After one day of incubation each flask was resuspended in 1.5 ml of medium and cells were seeded in 150 ml of medium in a T300 flask. After 48 days of growth at 37 °C in 5% $CO_2$, DNA of samples treated with 0.5, 1, 2 μg/ml of chloramphenicol was extracted. The samples of 15 μg/ml were processed after 72 additional hours. This experiment was repeated twice, and DNA samples were sequenced independently. Also, in parallel, *MPN-transformed* cells were spread on Hayflick agar plates supplemented with 20 μg/ml chloramphenicol and incubated at 37 °C in 5% $CO_2$. CFUs were accounted for after 1 week. The percentage of transformants was estimated by:

$$transformants[\%] = 100x \frac{CFUsHF + Cm}{CFUsHF} \qquad (1)$$

To test the smORF candidates, *MPN* cells were electroporated with 1ug of DNA. After 3 h we seeded the transformed cells in Hayflick agar plates supplemented with 2 μg/ml of tetracycline and either 5,10 or 15 μg/ml of chloramphenicol. CFUs were counted after 15 days growing at 37 °C in 5% $CO_2$.

### Transformation of mycoplasma agalactiae

We transformed *M. agalactiae* with a *BarnB* vector in a modified version of Tn4001 designed in a previous publication[38]. This includes an RBS motif upstream to the transposase to efficiently transform this organism, and another upstream to the chloramphenicol acetyltransferase cassette used for selecting transformed cells and found downstream the barnase gene. Transformations of *M. agalactiae* were performed as follows: cells were grown with shaking in 10 ml of Hayflick supplemented with 0.5% of sodium pyruvate. Prior to electroporation cells were pelleted (10 minutes at 4 °C at 10,000 g), washed 3 times with cold PBS, resuspended in 300 μl of electroporation buffer and disaggregated. 80 μl of cells were electroporated with 1ug of DNA and resuspended in a final volume of 500 μl by adding 420 μl of Hayflick +0.5% Sodium pyruvate. After a 90 minutes incubation at 37 °C, each transformation was seeded on Hayflick-0.5% pyruvate agar plates supplemented with either 100 μg/ml of gentamicin, 5, 10 or 15 μg/ml of chloramphenicol, or without any antibiotic, and incubated at 37 °C in 5% CO2. CFUs were counted after 5 days.

### Estimation of efficiencies of transformation in different libraries

As described above, the efficiencies of transformations shown in Fig. 1c were measured by counting the colony forming units (CFUs) in plates with and without the antibiotic and doing the ratio. The analysis of the variance was done from four different transformations (n = 4) and the different experiments were normalized versus one of the samples: $TnP_{438}catIR^*$ for the libraries of the experiment of chloramphenicol selection, $TnP_{Syn}eryIR^*$ for the libraries of the experiment of erythromycin selection and $TnP438catIR^*$ for the libraries of the barnase experiment.

### Sequencing of transposon libraries

Genomic DNA sequencing was performed in the Genomics facility at the Centre for Genomic Regulation in a HiSeq Sequencing v4 Chemistry controlled by Software HiSeq Control Software 2.2.58. Settings, 150 nucleotides in paired-end format. In the HiSeq Rapid Run sequencing technology from Illumina Genome Analyzer, the protocol starts with DNA fragmentation. Then, the fragmented DNA is amplified using oligos specific for the *cat*, *ereA* or barnase genes that also add adapters to the glass flow cell. Later, the sequencing is performed by synthesis cycles, in which a single complementary base

for each deoxynucleotide (dNTP) is incorporated using a fluorescently labeled dNTP. Finally, lasers excite the fluorophores while a camera captures images of the flow cell. In total, we sequenced 45 samples with 4 replicates for each *CmB5* and *CmB15* samples; 3 for each *CmB5A, Cm10A, Cm15A, Cm10B, Barnase*; and 2 replicates for the rest of the conditions presented, including *EryB1*. The raw data was submitted to the ArrayExpress database (https://www.ebi.ac.uk/biostudies/arrayexpress/studies/E-MTAB-10380) and assigned the accession identifier E-MTAB-10380.

### Definition of a *M. pneumoniae* annotation and intergenic database covering sequence features, -omics measures

We used the *MPN* M129 (NCBI Reference Sequence: NC_000912.1) genome sequence to define all putative ORFs, with translation product length ≥1 amino acids, from the six possible open reading frames (starts=ATG, TTG, GTG, stops=TAG, TAA). Considering *MPN* does not require ribosome binding sites (RBS) motifs to start translation[38], we did not set any size threshold as, theoretically, the resistance of the mutated transposon could be expressed in fusion with any translated sequence independently of its size. In total, 30,113 sequences were defined (Supplementary Data 4), these included the 690 known annotated coding sequences of *MPN*. For each sequence, all the available information was recapitulated, including coordinates, protein localization and function. We also included transcription-related information as to whether the annotation belonged to an operon or not, average expression (as log2(gene read count/gene length) and estimated average RNA copies per cell considering 4 RNA sequencing samples covering different growth times (6, 24 and 48 hours) available at ArrayExpress with the identifier -E-MTAB-6203. From previous studies, we considered the detection at protein and peptide level, available for 12,426 sequences that present an amino acid length ≥19 (from 116 MS experiments, ID PRIDE: PXD008243), average protein copies per cell, estimated half-life, and homology with a database including 109 smORFomes[10]. Finally, we also included transmembrane segment predictions and signal peptide presence estimated using TMHMM[67] and Phobius[55], respectively.

For the Ribosome Binding Site inclusion rate calculation, 15 bp upstream start codons we look for any of the motifs reported to act as Shine-Dalgarno motif: GGA, GAG, AGG, AGGA, GGAG, GAGG, AGGAG, GGAGG, AGAAGG, AGCAGG, AGGAGG, AGTAGG, AGGCGG, AGGGGG and AGGTGG[68].

The topology prediction by TMHMM consists in assigning the label *i*=cytoplasmic, *m*=membrane, *o*=outer to represent the location of the segment with respect to the membrane using predictions from with the tmhmm python library. To perform the different analyses, we reduce this information to the percentage of aa with the *i* label with respect to the total aa length. Besides, and to have a negative control in the analyses, we defined a set of intergenic sequences with their coordinates extracted from all the genome spans between ORFs distinguishing between strands and presenting low RNA expression profile with values log2(RNA read count/gene length) <2 (n = 1700). This set includes a total of 786 intergenic annotations extracted from the positive orientation (average sequence length = 25 ± 20 bp) and 914 from the negative orientation (24 ± 19 bp; Supplementary Data 6).

### Transposon insertion sites calling and visualization

We used FASTQINS[31] to retrieve the number of times, as read count, each base along the *MPN* genome (816,394 bp) was found next to a transposon insertion event. This was done using the following command 'fastqins -i <raw reads containing IR > -i2 <raw reads other pair > -g <genome > -t <IR sequence >'. This pipeline selects for reads, including specific sequences known as inverted repeats (IR) introduced during the transposition (Table 2), trims the inserted segment and maps the remaining sequence to the reference genome reporting the base pair position next to the trimmed section that corresponds to

**Table 2 | Inverted repeat sequences used to call strand-specific transposon insertions**

| Selection Marker | Sample type | in-frame selection | IR sequence passed to FASTQINS |
|---|---|---|---|
| Cm | A | No | CGAGGGGGGGCCCTTTTACACAATTATACGGACTTAATC |
| | B | Yes | TGATTTTTTTCTCTTTTACACAATTATACGGACTTAATC |
| | C | No | CGAGGGGGGGCCCTTTTACACAGTTGTACGGACTTAATC |
| | D | Yes | TGATTTTTTTCTCTTTTACACAGTTGTACGGACTTAATC |
| Barnase | B | No | TAACCTGTGCTTTTACACAATTATACGGACTTAATC |
| Ery | A | No | CTTATAATTTTACACAATTATACGGACTTAATC |
| | B | Yes | GTCAGGTTTTTACACAATTATACGGACTTAATC |

the insertion point. For this task, FASTQINS make use of standardized sequencing-processing tools Bowtie2 v2.5.2[69], SAMtools v1.18[70] and BEDTools[71] v2.31, managed by a Ruffus pipeline structure[72]. We consider our settings strict as we only consider reads mapping in paired-end, unambiguously and with no mismatches. As we were interested in extracting the orientation of the transposon insert, the IR sequence used to select reads was extended to include the beginning of the resistance/marker and FASTQINS was run using the strand-specific mode in the way the results for the positive and negative strand will include only insertions with the resistance/marker oriented in the positive or negative sense, thus producing viable fusions, respectively. After running this procedure over our library, including 39 samples, we obtained 78 profiles (one per each genome strand orientation) and the genome *coverage* (percentage of the genome that was found disrupted) and the total read count per sample (sum of read count for every position, Supplementary Data 2 and Supplementary Data 3). Visualization and main analysis of genome insertion profiles has been achieved using the plot function from ANUBIS[31] based on matplotlib[73] v3.8 and seaborn[74] v0.13 libraries, and scikit-learn v1.3 and SciPy[75] 1.11.3 for calculations. The repository https://github.com/samuelmiver/protinseq presents the required commands to extract the insertions from a raw sequencing file and the metrics used in this study in addition to basic plotting functions to allow the exploration of the profiles and produce the figures presented in this work.

**Genome-base level labeling to explore transposition selection**

Taking as reference the 30,113 ORFs found in *MPN* M129 and considering the design of our transposon where only insertions happening in the first position of a codon can produce viable fusions, we labeled each position in *MPN* genome with the following excluding labels: the first label, *annotated*, is assigned to bases corresponding to the first positions of codons in annotated proteins, thus, an insertion found there would express that protein in fusion with our selection resistance/marker. We assigned this label to the 17.4% ($n_{pos}$ = 142,443) and 12.5% ($n_{neg}$ = 102,840) of the positions in the positive and negative strands, respectively, of *MPN* (genome size = 816,394 bp). The second label, *putative*, is considered the same as *annotated* but taking only nonannotated entries. This covered 39.8% ($n_{pos}$ = 325,115 bp) and 44.1% ($n_{neg}$ = 360,302 bp) of the positive and negative genome strands. Finally, the *non-coding* label was assigned to the 42.7% ($n_{pos}$ = 348,836 bp) and 43.3% ($n_{neg}$ = 353,252 bp) of the positions, representing those cases where an insertion would be considered as inexplicable as no translation is expected. This last group includes, for example, second and third positions of codons in any annotation (if it does not present overlapping annotations) or any position located in-frame and downstream to a stop codon (this last case will correspond to positions within the intergenic annotation defined in the previous section). Additionally, we also considered within the *annotated* two different subsets of positions corresponding to in-frame positions of a set of genes with known E and NE essentialities, described as essentiality 'gold set' in *MPN*[29], including the following sizes $nE_{pos}$ = 5823 bp, $nE_{neg}$ = 10,139 bp, $nNE_{pos}$ = 4258 bp and $nNE_{neg}$ = 5823 bp. Labels and

the read count associated with each sample are included in Supplementary Data 3. For each of these positions types, we accounted for the *coverage* (percentage of positions found disrupted), total read count, the mean, median and standard deviation of read distributions under 4 different filtering conditions: no filter (0), removing 0-reads positions (1), ≥16 reads positions (16) and filtering out reads below the 5th percentile and above the 95th percentile (90) as suggested by previous transposon sequencing studies[76]. This information is included for each sample in Supplementary Data 4. Genomic and gene *coverage* and read count explorations were performed within the ANUBIS transposon sequencing exploration framework, which includes automated functions to retrieve these values from FASTQINS generated profiles[31].

**Identification analysis by ProTInSeq**

For each sample presenting a selective profile, we first filter out insertions with read values in the range of the tails of the *read counts* distribution and ignore repeated regions where mapping is inefficient as done in previous studies[76]. Distinguished by strand and replica, we model the background of the *coverage* distribution from *non-coding* positions with no RNA expression (log2(reads/bp)<2) in the *MPN* genome and we calculate the probability of each ORF to fit that distribution. Then, we consider as 'identified' those ORFs presenting a significant increase of insertions ($P < 0.05$), thus presenting a higher rate of in-frame insertions than expected by chance, normalized by their expected gene length. These evaluations were performed with the *Poisson* prediction method implemented in ANUBIS[31]. Insertion number and reads mapped to the ORFome of *MPN* and intergenic regions for the selective samples analyzed in this study can be found in Supplementary Data 7 (for selective samples considering in-frame insertions) while the same values mapped to whole genes and out-frame positions for all samples can be downloaded from Zenodo by the digital object identifier 10.5281/zenodo.7288780.

As we rely on gene transposon insertion *coverage* to estimate the statistic, this is sensitive to the inherent random nature of the transposon sequencing technique. Thus, to retrieve candidates with a higher number of insertions than expected by chance, we compared the annotated genes of *MPN* (Positives, P; $n$ = 690) against the set of negative control sequences derived from intergenic regions (Negatives, N; $n$ = 1700). Using a Receiver Operating Characteristic (ROC) we evaluated the relation between True Positive Rate as TP/(TP + FN) (i.e. true positive, or TP, for annotated protein detected; and false negatives, or FN, for annotated proteins with no signal), and False Positive Rate as FP/(FP + TN) (i.e. false positive, or FP, for intergenic annotations detected as ORF; and true negatives, or TN, for intergenic annotations with no signal). The Area Under the Curve (AUC) increases with high TPR and low FPR values; thus, it can be used to minimize the FPR and as a threshold to ensure all the candidates present more insertions than what could be expected by chance. In addition to this, we set a second condition for the detection, which requires an ORF to be reproducible in at least two samples (Supplementary Data 8 and Supplementary Fig. 4). Using this approach for each *CmB* and *CmD*

sample, we retrieved an average positive recall (i.e. the percentage of annotated proteins retrieved) of 77.6% ± 8.9% and a negative recall of 0.65% ± 0.37%, which corresponded to intergenic sequences in the negative control being detected with a signal like CDS. After three additional passages, we observed decreases for both the negative recall (from 0.65% to 0.43%) and number of identified proteins (from 60.4 to 34.8%) (Supplementary Data 8). A reduction in identified proteins is expected, as serial passages cause some NE/F genes to acquire resembled more essential profiles[31]. The putative false positives are derived from short non-coding sequences (<90 bp) that present a maximum of 5 insertions. Filtering them out by gene *coverage* value would exclude a wide range of E genes from the study (which only retain insertions at the N- and/or C-termini). To facilitate the analysis of these Tn-Seq mutated libraries, we have implemented new options to our previously published bioinformatic tools for essentiality studies. First, the pipeline of transposon insertions mapping (FASTQINS) includes a strand-specific mode to separate insertions by orientation. On the other hand, the set of essentiality assessment tools included in ANUBIS present new functions and subroutines to perform the different processing and estimation analyses distinguishing by frame and visualize this data[31].

As a proof of concept, we evaluated the number of insertions required to achieve similar identification results (highest selection conditions as in *CmB 15*, 75% of genes identified, assuming ~30% of them will be essential as observed in *MPN*) in 108 additional bacterial genomes with diverse genome size and number of genes annotated (Supplementary Data 12). We observed that in *E. coli* (5.2 Mb), a total number of 1.5 million unique insertions would be required to retrieve similar results. The highest number of unique insertions in this organism has been ~775,000[77], which is still far from the coverage observed in *MPN*. This coverage could be increased by combining multiple transformations to obtain comparable results. Nevertheless, SEPs could still be discovered in larger genomes when a transposon with enough efficiency is available. Furthermore, one could argue that the absence of RBS in MPN necessitates increased coverage to compensate for higher translational noise, a constraint that should not apply to other species.

### Identification analysis by ribosome profiling
Ribosome counts per base-pair (Supplementary Data 13), representing the number of times a ribosome is found binding an RNA in an exact genome position (discriminating by strand), were obtained for *MPN*, consisting of two biological replicates available at ArrayExpress under identifier E-MTAB-11935 (https://www.ebi.ac.uk/biostudies/arrayexpress/studies/E-MTAB-11935)[48]. In this publication, specific data procedures, such as discarding the first and last 15 bp of each annotation, were applied to overcome the complex signal obtained. We took into consideration the ribosome coverage (*RCV;* average of ribosome counts per base) estimated for each annotation with no discrimination between frames due to the limited selection observed for frame 1 (i.e., similar ribosome ratios per codon position, Supplementary Fig. 12; Supplementary Data 14). As a sanity check, we measured the correlation between the average *RCV* with the different frames and obtained Pearson correlation coefficient of 0.71, 0.84, and 0.88, for frames 1 to 3, respectively. Then, we calculated this same value for a background negative set consisting of intergenic annotations between operons, rRNAs, tRNAs, and ncRNAs, under the assumption they should not present ribosome footprints. Finally, a threshold was set in $RCV \geq 5$ as the number that retrieved less than 5% of the sequence annotations in the negative set. These same criteria were used to consider a smORF as translated and identifiable by Ribo-Seq.

### Definition of a curated set of SEPs in *Mycoplasma pneumoniae* and predicted functional properties
A curated union set of SEPs in *MPN* ($n = 302$, Supplementary Data 10) was established by combining experimentally identify SEPs by

ProTInSeq ($n = 153$), Ribo-Seq ($n = 95$) and MS ($n = 11$), in addition to sets predicted computationally by RanSEPs ($n = 118$)[10], and running BLASTP against the SmProt2 database[78] containing smORFs identified from literature, MS, Ribo-Seq and human microbiome studies. The tool smORFinder[79] was also tested, returning 6 SEP hits. However, all these corresponded to already annotated SEPs in *MPN* (MPN100, MPN204, MPN283, MPN410, MPN504, and MPN68).

These sets of SEPs were then submitted and inspected by different functional prediction servers (using default parameters, if not stated otherwise) and tools to provide an overview of the mechanisms SEPs could be involved in *MPN*. Specifically, we run i) an interspecies BLAST search against a previously defined collection of NCBI annotated SEPs[10]; ii) PfamScan to identify protein motifs[56]; iii) functional predictions based on orthology from EggNOG[59]; iv) fast suffix array neighborhood search to find homologous sequences in the UniProt database by PANNZER2[58]; v) signal peptide by SignalP and transmembrane motifs by Phobius[55]; vi) antimicrobial peptide (AMP) score by AMPred[54]; and structure-based and functional residue identification function prediction by DeepFRI[57]. Then, when a SEP was considered to have a predicted function when one of these conditions was satisfied: probability to be an AMP ≥ 75%, signal peptide predicted, hit with an annotated function in NCBI, Pfam motif identified, PANNZER2 description found, or a GO term for a molecular function is assigned by DeepFRI. A summary of these results can be found in Supplementary Fig. 14 and Supplementary Data 11.

### Reporting summary
Further information on research design is available in the Nature Portfolio Reporting Summary linked to this article.

## Data availability
The transposon sequencing raw data generated in this study have been deposited in the ArrayExpress database under accession code E-MTAB-10380, available in the following the link. The RNA-Seq raw sequencing files used in this study can be accessed in ArrayExpress under the identifier E-MTAB-6203 or accessing the project link. The ribosome profiling sequencing data for *M. pneumoniae* used in this study can be accessed in ArrayExpress under the identifier E-MTAB-11935, accessible using the link. The mass spectroscopy datasets for *M. pneumoniae* used in this study can be accessed in the PRIDE database under the identifier PXD008243 accessible using the link. The processed ProTInSeq data explored in this work, including base- and gene-level insertion mappings, can be found at Zenodo with the record ID 7288780 and Digital Object Identifier 10.5281/zenodo.7288779 accessible using the link https://zenodo.org/records/7288780. Source data are provided with this paper. All Supplementary Data descriptions included as Supplementary Notes in the Supplementary Information document. Source data are provided with this paper.

## Code availability
The software and code required to extract the insertion from raw reads and to analyze the insertions depending on the frame of insertions to given ORFs in a genome can be found in the repository https://github.com/samuelmiver/protinseq and citable using the DOI: 10.5281/zenodo.10637277 accessible using the link https://zenodo.org/records/10637277.

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

## Acknowledgements

We acknowledge the support provided by the Genomics Unit at the Centre for Genomic Regulation sequencing the samples. We also acknowledge Marc Weber for support in the analysis of Ribo-Seq data, and Veronica Raker for language editing. This project has received funding from the European Research Council (ERC) under the European Union's Horizon 2020 research and innovation programme MYCO-CHASSIS (670216) and  ERC LUNG-BIOREPAIR (101020135) ERC Advanced Grants (AdG). We also acknowledge support of the Spanish Ministry of Science and Innovation through the Centro de Excelencia Severo Ochoa (CEX2020-001049-S, MCIN/AEI /10.13039/501100011033), the Generalitat de Catalunya through the CERCA programme and to the EMBL partnership. We are grateful to the CRG Core Technologies Programme for their support and assistance in this work.

## Author contributions

S.M.V. performed computational and statistical analyses, interpreted results, created the figures and tables, and wrote the original draft. R.M., A.B., and C.S. performed library generation experiments and culture validation experiments. M.L.S. and L.S. provided direct supervision, funding acquisition, and were involved in the interpretation of results. S.M.V. and L.S. reviewed and edited the final manuscript version. S.M.V., M.L.S. and L.S. conceptualized the approach. All authors read and approved the final manuscript.

## Competing interests

The authors declare no competing interests. All collaborators in this study have met the authorship criteria required by the Nature Portfolio journals and have been included as authors. Roles and responsibilities were agreed among the collaborators prior to the research.
