## [Peer Review File · Nature Communications]

Reviewers' Comments:

Reviewer #1:

Remarks to the Author:

The manuscript "ProTINSeq: ultra-deep DNA sequencing applied to protein detection, quantification and functional studies" describes a tool adapted from transposon sequencing insertion (TIS) focused on proteome studies. By mutating transposon sequences, the authors successfully generate libraries of mutants where the selection markers express if inserted in-frame with native proteins. The technique's quality controls and analysis demonstrate its accuracy. The authors provide a detailed methodology that could be easily applied to different bacteria and includes adapted analysis pipelines. In conclusion, this work looks like a rigorous and well-developed methodology paper.

What is hard to envision from the actual version is how this technique would contribute to new exciting, relevant, or groundbreaking discoveries in bacterial functional genomics/proteomics. In the abstract, the authors propose that "ProTINSeq can be used to detect translational noise, for protein quantification and to provide insight into functional protein aspects such as relative half-life, stability, and membrane topology." What would be the advantages in terms of the scientific relevance of using ProTINSeq to identify when a bacterium has translational noise? It is not clear how to use this tool for protein quantification. The relative half-life and stability are only suggested based on some overlap with proteomics experiments performed with protease knockouts, but this aspect of MPN biology is not further studied. Does the determination of membrane topology increase the actual knowledge about membrane proteins? Does ProTINSeq make it easier to annotate/identify membrane proteins in a bacterial genome?

The introduction focuses primarily on SEPs, and the authors propose, although without independent experimental validations, that ProTINSeq successfully identifies several annotated SEPs plus five new >100 amino acid non-identified ORFs. But besides interspecies BLAST analysis, the identified SEPs would still need further investigation.

To conclude, this manuscript describes a reliable technique supported by substantial work. However, it needs a set of validation experiments, convincing arguments, or new ideas proving the usefulness of this technique to address biological questions of interest.

Minor comments

Page 5, lines 148-161: Authors need to clarify what they want to show by comparing the differences in coverage obtained when using different concentrations of chloramphenicol. With low antibiotic concentrations, tolerant wild-type strains could grow on plates causing contamination that could affect the coverage numbers depicted. Another hypothesis could be the proposed MPN "translational noise." Why don't the authors focus on the highest concentration showing enrichment in coding sequences for the B and D transposons? Also, the authors use coverage to describe various metrics. The repeated use of this term makes the text and figures axis tricky to understand. What coverage means in each case could be defined in a glossary as supplementary material.

Reviewer #2:

Remarks to the Author:

The manuscript described a deep sequencing strategy/methodology that provides a means to more reliably identify open reading frames including novel small expressed proteins. The method appears capable of determining their relative quantification, and provides deep insight into the determination of membrane topology. Using stringent criteria the authors claim identification of 75.2% of *M. pneumoniae* (MPN) proteome, which includes 66% of its annotated SEPs, 153 new smORFs, 5 new ORFs, and 11 targets of Lon protease. The latter is significant in that these proteins cannot be identified by mass spectrometry because of their stability and rapid degradation. Of the 158 newly identified proteins, we observe a wide variety of sizes range from 5 amino acids to 252 amino acids. Notably, of the 153 new SEPs, 54 were predicted using a previously published computation approach to the detection of SEPS.

The methodology appears to be robust and the authors have been rigorous in the design of controls strategy's to minimise false positive ORF identification. The authors have substantially increased the number of SEPs identified in the important human respiratory pathogen *Mycoplasma pneumoniae*. In addition the method highlights some fundamental new biology. The methodology is not only pertinent to the study of mycoplasmas but can be applied more broadly.

The manuscript is well written and clearly set out.

I have no suggestions for improvement.

Reviewer #3:

Remarks to the Author:

The manuscript "ProTInSeq: ultra-deep DNA sequencing applied to protein detection, quantification and functional studies" by Miravet-Verde et al. presents a method for characterising proteomes using transposons to detect translated ORFs. In some ways it can be described as an orthogonal method to mass spectrometry and ribosome profiling and is in principle interesting as an alternative approach to detect novel translated ORFs/SEPs/smORFs. However based on the data presented in the manuscript it seems the method has several weaknesses compared to these approaches and it is currently unclear to me why one would choose to use this method instead.

On the positive side it is clear that the method works in the context of this genome and can provide annotation of some novel ORFs. As a general method for peptide/smORF/SEP discovery, however, it appears unlikely to perform at a similar level as competing methods such as ribosome profiling. Ribosome profiling is mentioned in the introduction, but is superficially and quickly dismissed with the statement that it predicts peptides that "have no functional association". The proposed method, however, is also quite limited in addressing functional aspect besides having a bias against essential genes, leaving it open what the advantages of this approach would be. Unfortunately, there also appears to be additional disadvantages:

It is unclear whether this method would scale to bigger genomes or "any living system" as stated in the abstract. Methods like ribosome profiling and mass-spec essentially scale with the size of the translated "ORFome". This method instead scales with the size of the genome since insertions can occur anywhere. It is therefore not clear from the manuscript whether this method would be feasible on e.g. the human genome where the coding regions is less than 2% of the genome. The presented genome is extremely compact ("genome-reduced bacterium") with very little non-coding material.

The abundance estimates are likely less accurate than either ribosome profiling and mass-spec, both which give a more direct view into the amount of translation and the protein abundance respectively. Insertion also changes the size and content of the ORF and in likelihood the stability of the protein.

The method is biased in detection depending on the function of the genome. E.g. whether it is "essential" or whether it is a membrane protein. The latter is exploited to give insight into the topology of membrane proteins in an interesting twist, but ribosome profiling would appear to be a much less biased approach.

The method is based on altering the ORFs and therefore potentially changing the properties of the protein/SEP that it produces (e.g. stability as mentioned in #2).

A comparison of this method with the current state-of-the-art would have been prudent. A convincing case for the applicability of this method could potentially be made by:

- 1) comparing this approach with ribosome profiling, pointing out the advantages and/or demonstrating why this method should be used orthogonally. What are their respective accuracy and what number of sequenced reads/time/investment is necessary for the two approaches
- 2) showing that the method can indeed scale to larger genomes (e.g. a similar or lower number of

reads is necessary to achieve similar performance to ribosome profiling),
3) showing that quantification is comparable or better than these approaches (the comparison in the manuscript is mainly visual and hard to interpret in terms of how confident one can be in quantification).

In summary, the impression is that the method could potentially be used as an orthogonal assay to ribosome profiling to provide further information about small bacterial genomes, but if the limitations listed above are correct it is unclear how useful this would be and it seems unlikely to have a wider applicability. In the latter case, the language of the abstract and introduction oversells the method and should be adjusted to reflect the method's position relative to the state-of-the-art.

Minor:

The terms Ultra-sequencing, high-throughput sequencing, ultra-deep sequencing appear to be used interchangeably. It is not clear whether there is a difference in depth or not.

EDITOR COMMENTS

Dear Professor Serrano,

I apologize again for the delay in making a decision on your manuscript. Thank you again for submitting your manuscript "ProTInSeq: ultra-deep DNA sequencing applied to protein detection, quantification and functional studies" to Nature Communications. We have now received reports from 3 reviewers and, after careful consideration, we have decided to invite a major revision of the manuscript.

As you will see from the reports copied below, the reviewers raise important concerns. We find that these concerns limit the strength of the study, and therefore we ask you to address them with additional work. Without substantial revisions, we will be unlikely to send the paper back to review. **In particular, in agreement with Reviewers #1 and #3, we ask that you perform benchmarking and show the advantage of your method over existing methods, show that you can apply your method to larger genomes, show that you can get quantification similar to existing methods, and perform further validation.**

If you feel that you are able to comprehensively address all of the reviewers' concerns, please provide a point-by-point response to these comments along with your revision. Please show all changes in the manuscript text file with track changes or colour highlighting. If you are unable to address specific reviewer requests or find any points invalid, please explain why in the point-by-point response.

Reviewer #1 (Remarks to the Author):

The manuscript "*ProTInSeq: ultra-deep DNA sequencing applied to protein detection, quantification and functional studies*" describes a tool adapted from transposon sequencing insertion (TIS) focused on proteome studies. By mutating transposon sequences, the authors successfully generate libraries of mutants where the selection markers express if inserted in-frame with native proteins. The technique's quality controls and analysis demonstrate its accuracy. The authors provide a detailed methodology that could be easily applied to different bacteria and includes adapted analysis pipelines. In conclusion, this work looks like a rigorous and well-developed methodology paper.

We thank Referee 1 for the reviewing process, pointing out our effort in providing a valid methodology and demonstrating its accuracy. We have found Referee 1's comments very constructive and we have carefully addressed them. Below we describe the modifications made following the feedback provided.

What is hard to envision from the actual version is how this technique would contribute to new exciting, relevant, or groundbreaking discoveries in bacterial functional genomics/proteomics. In the abstract, the authors propose that "*ProTInSeq can be used to detect translational noise, for protein quantification and to provide insight into functional protein aspects such as relative half-life, stability, and membrane topology.*" What would be the advantages in terms of the scientific relevance of using ProTInSeq to identify when a bacterium has translational noise?

We extend our apologies for any prior ambiguity regarding the potential contributions of our methodology to the field. In response, we have now included in the introduction and discussion sections with supplementary references and text to underscore the rationale for investigating the extensive repertoire of small open reading frames (smORFs) within a bacterial context. Additionally, we emphasize the applicability of Tn-Seq, a widely employed technique for differential essentiality studies, as a valuable tool for examining conditionally-expressed small ORF-encoded proteins (SEPs) that may elude detection by alternative high-throughput methodologies.

Regarding translational noise, what we see is that for many SEPs not detected by mass spectroscopy or other methods, we identify them when fused in frame to the inserted positive marker. The reason probably is that they are very quickly degraded and upon fusion they are stabilized. This is supported by the fact that some regular ORFs that we could only detect if we downregulate the expression of Lon protease (from <https://pubmed.ncbi.nlm.nih.gov/33320415/>), are detected by ProTInSeq. Our analysis shows that there is a significant amount of potential SEPs that are translated and quickly degraded (Translational noise) that will be difficult to detect by other methods. This new manuscript version highlights this point and the suggested implications of this effect in protein complexity, adaptation, or to mitigate transcriptional noise (last sentence in the third paragraph of the introduction). In addition, we have extended the discussion (see second paragraph) with the possible implications of this noise in *M. pneumoniae* growth and/or helping to mitigate the effect of its significant transcriptional noise; and ProTInSeq as an approach to discriminate translational and technical noise in cases where Ribo-Seq and mass spectroscopy results do not converge (see new Extended Data Figures 5 and 12).

It is not clear how to use this tool for protein quantification. The relative half-life and stability are only suggested based on some overlap with proteomics experiments performed with protease knockouts, but this aspect of MPN biology is not further studied.

We analyzed the correlation with protein abundances measured in 116 mass spectroscopy experiments and showed that ProTInSeq's gene insertion coverage at NE regions correlates well with protein abundance bins. Although not comparable with ribosome profiling approaches for E regions (see the last paragraph in the results section 'Factors affecting protein identification by ProTInSeq and its use for relative quantification of expressed proteins'), the correlation is comparable to ribosome profiling for NE genes. As suggested by reviewer 3 (below), the language and conclusion regarding protein quantification have been toned down to acknowledge that the main purpose of the method is not quantifying although as a bonus good numbers can be obtained for NE proteins with the same experimental setup used to identify SEPs.

Regarding the protease knockout paper mentioned here, this was used to highlight how the proposed method can identify elusive proteins fastly degraded in the cell thanks to the fusion to the resistance gene.

Does the determination of membrane topology increase the actual knowledge about membrane proteins? Does ProTINSeq make it easier to annotate/identify membrane proteins in a bacterial genome?

As in the previous point, we have toned down the language to acknowledge that the main purpose of the method is not predicting membrane topology. However, ProTInSeq can be used to predict membrane topology for F and NE proteins and is useful in validating conflicting predictions between topology prediction softwares as shown with the example of MPN359 (Fig 4c). For E membrane proteins it can be used to detect the beginning of the first transmembrane segment or if the C-termini of the protein is in the cytoplasm.

The introduction focuses primarily on SEPs, and the authors propose, although without independent experimental validations, that ProTINSeq successfully identifies several annotated SEPs plus five new >100 aminoacid non-identified ORFs. But besides interspecies BLAST analysis, the identified SEPs would still need further investigation.

It is important to indicate that ProTInSeq independently identifies 3 SEPs previously identified by mass spectroscopy and 5 using C₁₃ labeled peptides (<https://doi.org/10.15252/msb.20188290>), as well as 22 annotated SEPs in *M. pneumoniae* and 54 previously predicted by the computational approach RanSEPs.

Taking into account the reviewer comments in our new manuscript version we present extended results obtained with a pair of independent ribosome profiling samples (see response to reviewer 3). On the

functional aspect and in addition to the results retrieved by interspecies BLAST and antimicrobial and signal peptide prediction, we have now collected functional predictions from EggNOG (based on orthology), DeepFRI (structure-based and functional residue identification function prediction), PANNZER2 (fast suffix array neighborhood search (SANSparallel) to find homologous sequences in the UniProt database), and PfamScan (Pfam HMM). This analysis provides evidence for function for 50 SEPs identified by ProTInSeq and 24 by Ribo-Seq. The methodology followed is included in the section ‘Function exploration and validation of new SEPs by computational and experimental analyses’, and in the results section specific for new SEPs identification. Prediction of protein function is still challenging, even more so in SEPs where there are less studied cases, and we acknowledge this in the Discussion section. Notwithstanding, ProTInSeq retrieved a larger number of SEPs with predicted function (32%) compared to Ribo-Seq (25%) after running this analysis showing that our approach can identify high-value SEPs targets for further specific inspection.

To conclude, this manuscript describes a reliable technique supported by substantial work. However, it needs a set of validation experiments, convincing arguments, or new ideas proving the usefulness of this technique to address biological questions of interest.

We appreciate the acknowledgment of the effort invested in this work and the constructive comments provided. As additional independent validation, we have now included a comparative with ribosome profiling experiments (see response to reviewer 3). Finally, we have tried to state better the usefulness and application of ProTInSeq as an orthogonal method to identify expressed SEPs and elusive proteins (*e.g.*, Lon protease targets) at the same time it provides insights in proteome characteristics using a flexible and cost-effective DNA-sequencing approach.

Minor comments

Page 5, lines 148-161: Authors need to clarify what they want to show by comparing the differences in coverage obtained when using different concentrations of chloramphenicol. With low antibiotic concentrations, tolerant wild-type strains could grow on plates causing contamination that could affect the coverage numbers depicted. Another hypothesis could be the proposed MPN “translational noise.” Why don’t the authors focus on the highest concentration showing enrichment in coding sequences for the B and D transposons?

The comparative study between antibiotic concentrations was performed to define the best concentration required to observe a significant selection when comparing in-frame insertions in annotated genes and using purely non-coding positions as negative control. On one hand, we wanted to identify the minimum concentration required to get a significant selection of insertions in-frame to a known gene to ensure maximum recovery of identified genes as higher concentrations will imply less insertions recovered. Although selection was already observed at 2 $\mu\text{g/ml}$ of chloramphenicol, only samples with 5, 10, and 15 $\mu\text{g/ml}$ were considered in further analyses. On the other hand, we observed that insertion coverage in *non-coding* positions in libraries B and D plateau at 1 insertion every 100 bp after 5 $\mu\text{g/ml}$ (technical noise, right panel in figure 2a); thus suggesting ≥ 5 $\mu\text{g/ml}$ concentrations share similar selection pressures. We have included this reasoning in the second paragraph of the second results sections. As stated in the second and third results sections, we only considered samples grown with ≥ 5 $\mu\text{g/ml}$ of chloramphenicol, with similar selection patterns, for further analysis and did not take into account lower concentrations that could be noisier as rightly pointed by the reviewer. Also, it allows us to estimate, as mentioned above, protein quantification for F and NE proteins since very abundant proteins will have insertions at high chloramphenicol concentration and the opposite is true.

Finally, we acknowledge the effect that translational noise could have in this comparative. To deal with this, we relied on stringent criteria based on read count, number of insertions per gene, and reproducibility (see Identification analysis by ProTInSeq in Methods). Furthermore, the second paragraph of the discussion has been extended to acknowledge the difficulty in discriminating between peptides produced by translational noise and actual SEPs functionally relevant for the cell.

Also, the authors use coverage to describe various metrics. The repeated use of this term makes the text and figures axis tricky to understand. What coverage means in each case could be defined in a glossary as supplementary material.

We apologize for the possible inconveniences in the terminology. The term *coverage* is widely used in transposon-based studies to refer to the number of insertions found in a genomic region normalized by the base-pairs length of that region. This metric can be used to describe rates of insertions at different genomic levels such as the ones explored in this article, which include genome and gene coverage referring to number of insertions normalized by genome size or by annotation length, respectively. We have clarified the meaning of the term when the concept is first introduced and in the methods, and ensured it is clear when we refer to genome or gene coverage. Finally, to highlight when this specific metric is used, coverage appears in italics when mentioned.

Reviewer #2 (Remarks to the Author):

The manuscript described a deep sequencing strategy/methodology that provides a means to more reliably identify open reading frames including novel small expressed proteins. The method appears capable of determining their relative quantification, and provides deep insight into the determination of membrane topology. Using stringent criteria the authors claim identification of 75.2% of *M. pneumoniae* (MPN) proteome, which includes 66% of its annotated SEPs, 153 new smORFs, 5 new ORFs, and 11 targets of Lon protease. The latter is significant in that these proteins cannot be identified by mass spectrometry because of their stability and rapid degradation. Of the 158 newly identified proteins, we observe a wide variety of sizes range from 5 amino acids to 252 amino acids. Notably, of the 153 new SEPs, 54 were predicted using a previously published computation approach to the detection of SEPS.

The methodology appears to be robust and the authors have been rigorous in the design of controls strategy's to minimise false positive ORF identification. The authors have substantially increased the number of SEPs identified in the important human respiratory pathogen *Mycoplasma pneumoniae*. In addition the method highlights some fundamental new biology. The methodology is not only pertinent to the study of mycoplasmas but can be applied more broadly.

The manuscript is well written and clearly set out.

I have no suggestions for improvement.

We sincerely thank Referee 2 for the reviewing process of our work and we acknowledge their positive opinion on the results obtained and how they are described. Specially, we appreciate the acknowledgement on how we have described and our taken strategies to minimize false positive SEPs identification.

Reviewer #3 (Remarks to the Author):

The manuscript "ProTInSeq: ultra-deep DNA sequencing applied to protein detection, quantification and functional studies" by Miravet-Verde et al. presents a method for characterising proteomes using transposons to detect translated ORFs. In some ways it can be described as an orthogonal method to mass spectrometry and ribosome profiling and is in principle interesting as an alternative approach to detect novel translated ORFs/SEPs/smORFs. However, based on the data presented in the manuscript it seems the method has several weaknesses compared to these approaches and it is currently unclear to me why one would choose to use this method instead. On the positive side it is clear that the method works in the context of this genome and can provide annotations of some novel ORFs.

We express our gratitude to Referee 3 for their valuable contributions during the review process, and we have carefully considered and addressed their constructive comments in this revised version. Additionally, we sincerely apologize for any ambiguity regarding the potential contributions of our methodology and application scenarios in the previous version. We have taken steps to enhance clarity in both the introduction and discussion sections. In this updated version, we explicitly recognize our method as orthogonal to mass spectrometry and ribosome profiling approaches for identifying SEPs. Simultaneously, we emphasize that our method offers supplementary information about a proteome within the same experimental framework, as highlighted in the concluding paragraph of the introduction.

As a general method for peptide/smORF/SEP discovery, however, it appears unlikely to perform at a similar level as competing methods such as ribosome profiling. Ribosome profiling is mentioned in the introduction, but is superficially and quickly dismissed with the statement that it predicts peptides that “have no functional association”.

We apologize for the dismissal of ribosome profiling application for the identification of SEPs. Current version of the article integrates with our results the comparative with the two ribosome profiling samples in *Mycoplasma pneumoniae* presented in E-MTAB-11935. Together with the already presented mass spectroscopy-detected SEPs, we believe the manuscript now provides a better overview of the different alternatives and the specific advantages of applying ProTInSeq for bacterial SEPs identification. For instance, in identifying overlapping SEPs with limited in-frame signal detection by Ribo-Seq in *Mycoplasma pneumoniae* (Extended Data Figure 11), or validating the expression of SEPs not identifiable by Ribo-Seq and mass spectroscopy.

The proposed method, however, is also quite limited in addressing functional aspects besides having a bias against essential genes, leaving it open what the advantages of this approach would be.

This method like Ribo-Seq or mass spectroscopy are not intended to provide biological evidence of the function of a protein. We apologize since we initially included the term ‘functional’ to refer to the relative quantification, essentiality, stability and topological (membrane proteins and lipoproteins) aspects of proteins that can be explored with ProTInSeq. Thus the title of the manuscript has been updated to better reflect the application of the method. The introduction, results and discussion have been toned down regarding functional aspects as well.

Regarding advantages, this method provides in one experiment a full essentiality analysis of all genes and non coding regions of a genome, identifies standard and small protein-encoding ORFs (including those that are unstable and quickly degraded and some missed by ribosome profiling and or mass spectroscopy) and therefore offers a vision on translational noise and can provide qualitative information on protein expression levels as well as information on membrane topology of F and NE proteins. Thus although complementary to ribosome profiling and or mass spectroscopy it offers some advantages.

Unfortunately, there also appears to be additional disadvantages:

It is unclear whether this method would scale to bigger genomes or “any living system” as stated in the abstract. Methods like ribosome profiling and mass-spec essentially scale with the size of the translated “ORFome”. This method instead scales with the size of the genome since insertions can occur anywhere. It is therefore not clear from the manuscript whether this method would be feasible on e.g. the human genome where the coding regions is less than 2% of the genome. The presented genome is extremely compact (“genome-reduced bacterium”) with very little non-coding material.

The referee is right that this method requires significant coverage to identify SEPs in larger genomes. We have toned down the last sentence of the abstract and it does not mention ‘any living system’ now. Furthermore, the discussion now includes the mentioned escalation limitations.

It is difficult to know the scale of ORFs translated in the human genome. In fact there is a debate to what extent long non coding RNAs encode for small proteins, and SEPs are found in every living organism. In addition, despite the consideration of *M. pneumoniae* as a genome-reduced bacterium, when considering known gene annotations, only 30% of the genes are essential thus difficult to detect by ProTInSeq. Taking this as reference and the observed ~30% insertion coverage in the highest selection conditions with the control libraries, we have estimated the number of insertions required to achieve similar identification results in 108 additional bacterial genomes with diverse genome size and number of genes annotated. For example, we observed that in *Escherichia coli* (5.2 Mb), a total number of 1.5 million unique insertions would be required to retrieve similar results. The highest number of unique insertions in this organism has been ~775,000 (<https://pubmed.ncbi.nlm.nih.gov/29339415/>) which is still far from the coverage observed in *M. pneumoniae*. This coverage could be increased by combining multiple transformations to obtain comparable results. Nevertheless, SEPs could still be discovered in larger genomes when a transposon with enough efficiency is available. Furthermore, one could argue that the absence of RBS in *Mycoplasma pneumoniae* necessitates increased coverage to compensate for higher translational noise, a constraint that should not apply to other species. These observations are acknowledged in the new version of the manuscript.

The abundance estimates are likely less accurate than either ribosome profiling and mass-spec, both which give a more direct view into the amount of translation and the protein abundance respectively. Insertion also changes the size and content of the ORF and in likelihood the stability of the protein.

We agree with Reviewer 3 on the limitation of our methodology and acknowledge our method is not comparable to ribosome profiling in terms of protein abundance quantification because of the E proteins. These observations, supported by new data, are now clearly stated in the results sections (see responses below). However, both methods of ribosome profiling and ProTInSeq have the same problem, the importance of protein half life. In Ribo-Seq one could overestimate protein abundance when a protein has a very short half life and in our method one could overestimate it if the genetic marker fusion can affect the stability of the protein, which is in fact the reason why we can detect elusive proteins with signals only when the Lon protease is knocked-out. ProTInSeq does not try to replace methods such as Ribo-Seq and mass spectroscopy but provides an additional way to identify and relatively quantify proteins that might be degraded quickly or overlapping a larger gene. Comparison of Ribo-Seq with ProTInSeq could serve to identify NE and F proteins with short half lives.

Thus the information provided by our approach is valuable considering the flexibility and cost-effectiveness in the application of Tn-Seq compared to alternative approaches.

The method is biased in detection depending on the function of the genome. E.g. whether it is “essential” or whether it is a membrane protein. The latter is exploited to give insight into the topology of membrane proteins in an interesting twist, but ribosome profiling would appear to be a much less biased approach.

The biases of the approach are extensively reported now in comparison to Ribo-Seq in the section ‘Factors affecting protein identification by ProTInSeq and its use for relative quantification of expressed proteins’. We have toned down the language to acknowledge the main purpose of the method is not quantifying or predicting membrane topology although these are features that can be explored with the same experimental setup used to identify SEPs. It is not obvious to us how ribosome profiling will inform on lipoproteins or on protein membrane topology.

The method is based on altering the ORFs and therefore potentially changing the properties of the protein/SEP that it produces (e.g. stability as mentioned in #2). A comparison of this method with the current state-of-the-art would have been prudent. A convincing case for the applicability of this method could potentially be made by:

- 1) comparing this approach with ribosome profiling, pointing out the advantages and/or demonstrating why this method should be used orthogonally. What are their respective accuracy and what number of

sequenced reads/time/investment is necessary for the two approaches 2) showing that the method can indeed scale to larger genomes (e.g. a similar or lower number of reads is necessary to achieve similar performance to ribosome profiling),

Current version of the article now integrates a full comparative with ribosome profiling and mass spectroscopy highlighting SEPs that can only be identified when using ProTInSeq. The comparative analyses for identification of standard proteins (>100 aa) are reflected in the last paragraph of the new section 'Benchmark of ProTInSeq and comparative with other experimental methods in identifying translation of annotated proteins' and at the end of the results section 'Factors affecting protein identification by ProTInSeq and its use for relative quantification of expressed proteins' were the correlation with protein abundances is evaluated between different techniques.

For SEPs, the identification capabilities of ProTInSeq in comparison to Ribo-Seq is now included in the last part of the results section 'Identification of new proteins and SEPs using ProTInSeq' and in 'Function exploration and validation of new SEPs by computational and experimental analyses'. This analysis highlights the benefits of using ProTInSeq (65 SEPs uniquely identified) as orthogonal to Ribo-Seq and MS method. In addition, ProTInSeq shows a higher a larger overlap with computational predicted methods, and a higher number of SEPs with a predicted functional property (35% for ProTInSeq hits, 25% for Ribo-Seq hits) when explored by several bioinformatic approaches (EggNOG, PANNZER2, BLASTP, DeepFRI). These results are summarized in the upset plots presented in Extended Data Figure 12 and Extended Data Figure 13.

Finally, we believe the application of this methodology has two additional advantages: (i) the possibility to identify SEPs overlapping other genes for which Ribo-Seq signal will be difficult to deconvolute from the one for the larger gene; (ii) providing a relatively cost-effective and less laborious methodology to identify bacterial SEPs when a transposon is available with sufficient transformation efficiency.

Following the reviewer suggestion, we have explored the number of reads required to get the results presented in the identification of (annotated) proteins and SEPs results section. Results, summarized in Extended Data Figure 6, show that less total number of reads sequenced and less recovered transposon insertions than ribosome footprints are required to achieve similar performance using ProTInSeq.

3) showing that quantification is comparable or better than these approaches (the comparison in the manuscript is mainly visual and hard to interpret in terms of how confident one can be in quantification).

The current version of the manuscript explores the relative quantification capacities in terms of protein abundance measured by mass spectroscopy for 560 proteins detectable in *Mycoplasma pneumoniae*. While Ribo-Seq presents a better correlation coefficient when taking all the genes into consideration, we show that ProTInSeq is comparable to ribosome profiling for F and NE genes. These results close the new section 'Factors affecting protein identification by ProTInSeq and its use for relative quantification of expressed proteins' and are recapitulated in the Extended Data Figure 8.

In summary, the impression is that the method could potentially be used as an orthogonal assay to ribosome profiling to provide further information about small bacterial genomes, but if the limitations listed above are correct it is unclear how useful this would be and it seems unlikely to have a wider applicability. In the latter case, the language of the abstract an introduction oversells the method and should be adjusted to reflect the methods position relative to the state-of-the-art.

After conducting a comparative analysis with Ribo-Seq and identifying novel SEPs that lack significant ribosome footprints, we are confident that ProTInSeq can offer valuable insights into the expression of SEPs from a bacterial genome. The current version of the manuscript has been revised to mitigate any potential exaggeration or overselling of the method's capabilities. We value the perspective

provided by reviewer 3 and acknowledge the suggested amendments, which, once implemented, result in a more robust and equitable presentation of our work.

Minor:

The terms Ultra-sequencing, high-throughput sequencing, ultra-deep sequencing appears to be used interchangeably. It is not clear whether there is a difference in depth or not.

We apologize for the confusion in the terminology. High-throughput sequencing refers to the general category of sequencing genetic material, while ultra-deep sequencing is a specialized approach within high-throughput sequencing that involves sequencing to an exceptionally high depth for specific genomic regions, in this case, transposon insertions and their genomic DNA neighboring regions. We have ensured that now the ProTInSeq method is referred to every time as an ultra-deep sequencing approach and included a clarification regarding its relation to high-throughput sequencing (Introduction; second paragraph). The term ‘ultra-sequencing’ in the abstract corresponded to typographical error now corrected.

Reviewers' Comments:

Reviewer #1:

Remarks to the Author:

The current version focuses more on SEPs and tunes down the usefulness of the technique for other purposes (protein quantification, protein localization) in a more realistic way. It would be interesting to see the application of this technique in other species to see if the SEPs identified here are indeed conserved.

Comments

"Moreover, beyond determining essentiality..."

Since gene expression level and location affect ProTInSeq results, low-expressed genes and exposed/secreted proteins coding genes could be wrongly assigned as essentials. The technique does not improve standard TnSeq methodologies to determine essential genes. This sentence proposes the technique as a good method to assess gene essentiality.

"Given secreted SEPs function in communication and competition in microbial ecosystems 39,40 , and the flexibility and cost-effectiveness of Tn-Seq, ProgTInSeq can validate the expression of elusive SEPs of relevance in microbial physiology 41 , help tackle the looming antibiotic resistance crisis 42 , and broaden the scope of biotechnological innovation."

Like many other parts of the previous version, this sentence oversells the impact of this work. ProTInSeq provides a complementary approach to other technologies for detecting SEPs. ProTInSeq, for what it is presented in this work, does not contribute to tackling the looming antibiotic resistance crisis and broadening the scope of biotechnological innovation.

"We transformed *M. agalactiae* with a BarnB vector in a modified version of Tn4001 33 , which includes a RBS motif, to efficiently transform this organism."

Does this mean the marker does not need to be in-frame to be expressed in this transposon version? It's hard for me to understand why this experiment supports the presence of translational noise in MPN. How was the RBS included? I am not finding that information in the material and methods section. Also, please note that the description of the Barnase transposon in the Supplementary data indicates the in-frame insertion with the cat, not the bar gene.

Reviewer #3:

Remarks to the Author:

All my concerns have been addressed.

Response to reviewers

Reviewer #1 (Remarks to the Author):

The current version focuses more on SEPs and tunes down the usefulness of the technique for other purposes (protein quantification, protein localization) in a more realistic way. It would be interesting to see the application of this technique in other species to see if the SEPs identified here are indeed conserved.

We sincerely thank Referee 1 for the reviewing process of our work and we acknowledge their positive opinion on the last version of the manuscript. We have addressed the final comments, which, like the previous ones, we found highly constructive and valuable.

We agree that the methodology presented would be interesting to apply to other species. Currently, we are working on establishing the method in two species with larger genomes than the one originally presented (*Mycoplasma agalactiae* and *Mycoplasma gallisepticum*, both having genome sizes >1Mb). The preliminary results observed indicate that in-frame selection can also be achieved in these cases.

Still, as reported in the start of the section ‘Function exploration and validation of new SEPs by computational and experimental analyses’ we explored the presence of SEPs identified using a BLASTP search against a database of 9,464,640 translated smORFs from 109 bacterial species obtained in the study Miravet-Verde, S. *et al.* (2019) <https://www.embopress.org/doi/full/10.15252/msb.20188290>. This reveals that out of the SEPs identified in this 178 study are present in at least 2 bacterial species. When excluding translated smORFs from closely-related species from the Mollicutes class, a total of 41 SEPs still present a hit in at least 2 evolutionary-distant species.

Comments

“Moreover, beyond determining essentiality...”

Since gene expression level and location affect ProTInSeq results, low-expressed genes and exposed/secreted proteins coding genes could be wrongly assigned as essentials. The technique does not improve standard TnSeq methodologies to determine essential genes. This sentence proposes the technique as a good method to assess gene essentiality.

We acknowledge that the proposed technique does not enhance standard TnSeq approaches in determining gene essentiality. Although the impact of mentioned factors (e.g., expression, exposure/secretion) is detailed in the results section titled ‘Factors affecting protein identification by ProTInSeq and its use for relative quantification of expressed proteins’, we still observe a correlation between the expected essentiality and the number of insertions found in frame with ProTInSeq (as illustrated in Figure 2b).

To further substantiate this observation, we calculated the correlation between normalized gene coverages obtained in the selective libraries transformed with the control and mutated chloramphenicol acetyltransferase at concentrations of 5, 10, and 15 µg/ml of chloramphenicol. This analysis reveals that: i) a high reproducibility is evident between concentrations within control and mutated samples, and ii) a lower correlation between control and mutated, though still significant. This is depicted in the newly added Supplementary Figure 2. Nevertheless, to prevent any potential misunderstanding, we have removed the mentioned sentence from the abstract.

“Given secreted SEPs function in communication and competition in microbial ecosystems 39,40 , and the flexibility and cost-effectiveness of Tn-Seq, ProTInSeq can validate the expression of elusive SEPs of relevance in microbial physiology 41 , help tackle the looming antibiotic resistance crisis 42 , and broaden the scope of biotechnological innovation.”

Like many other parts of the previous version, this sentence oversells the impact of this work. ProTInSeq provides a complementary approach to other technologies for detecting SEPs. ProTInSeq, for what it is presented in this work, does not contribute to tackling the looming antibiotic resistance crisis and broadening the scope of biotechnological innovation.

We apologize for the inaccurate wording, we were not pretending to suggest ProTInSeq directly contributes to tackle antibiotic resistances or broadening biotechnological innovation by itself but by the identification and validation of SEPs that in one quarter of the reported examples present antimicrobial or signal peptide potential. To avoid any misunderstanding, we have reworded the last sentence of the introduction and discussion sections that now say:

‘Given the relevant described functions of SEPs, including communication and competition in microbial ecosystems, and the flexibility and cost-effectiveness of Tn-Seq, we envision ProTInSeq as a valuable tool to validate the expression of elusive SEPs of potential relevance in microbial physiology’.

‘In conclusion, ProTInSeq can be used as an orthogonal method to identify expressed SEPs and elusive proteins at the same time it provides insights in proteome characteristics using a flexible and cost-effective DNA-sequencing approach’.

“We transformed *M. agalactiae* with a BarnB vector in a modified version of Tn4001 33, which includes a RBS motif, to efficiently transform this organism.”

Does this mean the marker does not need to be in-frame to be expressed in this transposon version? It’s hard for me to understand why this experiment supports the presence of translational noise in MPN. How was the RBS included? I am not finding that information in the material and methods section. Also, please note that the description of the Barnase transposon in the Supplementary data indicates the in-frame insertion with the cat, not the bar gene.

We extend our apologies for the confusing wording when describing the transposon used in *M. agalactiae*. In this case, the RBS is present in the additional chloramphenicol acetyltransferase cassette added downstream to the barnase cassette and used for selecting transformed cells. Opposite to *M. pneumoniae*, *M. agalactiae* needs a Shine-Dalgarno sequence close to the start codon. The definition of this modified mini-transposon to include an RBS has been published and referenced in the article Montero-Blay, A. *et al.* (2019). Within this transposon, the barnase gene still lacks the start codon, thus only expressed when inserted in-frame. Thus we will recover insertions only when they occur out of frame (shown in Supplementary Fig. 3).

Our reasoning to suggest this experiment supports MPN translational noise comes from the fact that a higher transformation efficiency is observed when using the barnase modified transposon in *M. agalactiae* (Fig. 2d) suggesting that a larger number of insertions in MPN produce a functional fusion of barnase producing the cell death. MPN does not require a RBS before the first start codon of the transcript (26% of the genes present them), while this is not the case for *M. agalactiae* (73% of genes with RBS). Thus transcriptional noise will result in increased translational noise in MPN compared to *M. agalactiae* since in MPN the first ATG in the transcript will be translated, while in *M. agalactiae* it will need a Shine-Dalgarno motif prior to the ATG. As a result and because both bacteria have similar numbers of conventional ORFs there will be more viable clones with barnase insertions in *M. agalactiae* as insertions in non-coding regions will be less likely to be translated than in MPN.

Finally, we appreciate the thorough revision of the supplementary material. As pointed out by Reviewer #1, we erroneously referred to chloramphenicol acetyltransferase instead of RNase barnase in the supplementary data when describing the design of the vectors. This has now been corrected.

Reviewer #3 (Remarks to the Author):

All my concerns have been addressed.

We sincerely thank Reviewer #3 for the whole reviewing process of our work and we acknowledge their positive opinion on the last version.